# PGC: Peak-Guided Calibration for Generalizable AI-Generated Image Detection

Xiaoyu Zhou [1]    Jianwei Fei [2]    Peipeng Yu [1]    Jingchang Xie [3]    Chong Cheng [1]    Zhihua Xia [1]

## Abstract

The rapid evolution of generative AI, from GANs to modern diffusion models, has resulted in increasingly subtle discriminative clues. These fine-grained signals are often overshadowed by dominant, high-fidelity image content (e.g., the main subject), limiting the reliability of existing detectors that predominantly rely on global representations. To address this challenge, we propose the **Peak-Guided Calibration (PGC)** framework. PGC introduces a novel strategy that aggregates salient features via a peak-focusing mechanism. Specifically, by employing a peak-sensitive aggregation that accentuates the most discriminative local clues, PGC leverages these critical signals to calibrate the global decision. This approach recovers subtle patterns that would otherwise be submerged in the global context. Furthermore, to better simulate real-world threats, we introduce the **CommGen15** dataset, a challenging benchmark comprising samples from 15 commercial models. Extensive experiments demonstrate that PGC achieves state-of-the-art performance. Specifically, it improves mean accuracy by **+12.3%** on our CommGen15 dataset, and sets new records on standard benchmarks, including GenImage (**+2.1%**), AIGI (**+3.5%**), and UniversalFakeDetect (**+1.7%**). Code is available at https://github.com/xiaoyu6868/PGC.

## 1. Introduction

The domain of AI-generated images has undergone a transformative shift, moving from the era of GANs (Goodfellow et al., 2014) to the modern diffusion models (Esser et al.,

[1]College of Cyber Security, Jinan University, Guangzhou, China [2]Department of Information Engineering, University of Florence, Florence, Italy [3]School of Integrated Circuits, Guangdong University of Technology, Guangzhou, China. Correspondence to: Zhihua Xia <xia_zhihua@163.com>.

*Proceedings of the 43rd International Conference on Machine Learning*, Seoul, South Korea. PMLR 306, 2026. Copyright 2026 by the author(s).

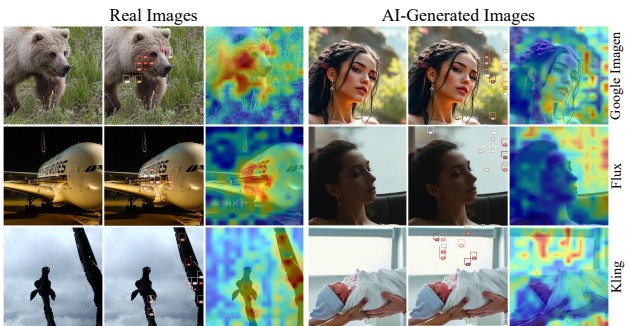

*Figure 1.* **Motivation.** Visualization of discriminative traces. Columns from left to right display: (1) The original input images; (2) Critical evidentiary patches identified by our PGC (highlighted with red bounding boxes); and (3) The decision heatmaps (CAM) of the PGC detector. **Observation:** While real images trigger attention on the main subject, AI-generated images (Google Imagen, Flux, Kling) shift focus to the background, indicating that the high-fidelity generation of the semantic subject overshadows the subtle discriminative artifacts.

2024). While early generative models left observable fingerprints, recent high-fidelity generators produce content with exceptional visual quality (Zhang et al., 2024). Although numerous detection strategies have been proposed, including low-level artifacts in the spatial (Tan et al., 2024b) or frequency domains (Tan et al., 2024a), and methods leveraging pre-trained foundation models (Liu et al., 2024; Ojha et al., 2023), performance degrades when confronting the state-of-the-art commercial forgeries (Guillaro et al., 2025).

This performance degradation stems from the spatially uneven allocation of generative attention. Unlike early models with globally consistent artifacts, modern algorithms prioritize the main semantic subject (Miao et al., 2025; Zhang et al., 2024), concentrating their generative capacity on foreground fidelity. This attentional bias creates a resource trade-off that rigorously optimizes the subject, effectively displacing discriminative artifacts to the less constrained background regions. However, existing detectors primarily rely on global representations (Tan et al., 2023; 2024b), which are biased toward high-energy semantic regions. This causes subtle, localized traces to be overshadowed by the dominant high-fidelity foreground. As illustrated in Figure 1, this semantic dominance results in a divergence in attentional patterns: while detectors focus on the semantic foreground (e.g., the main subject) in real images, their

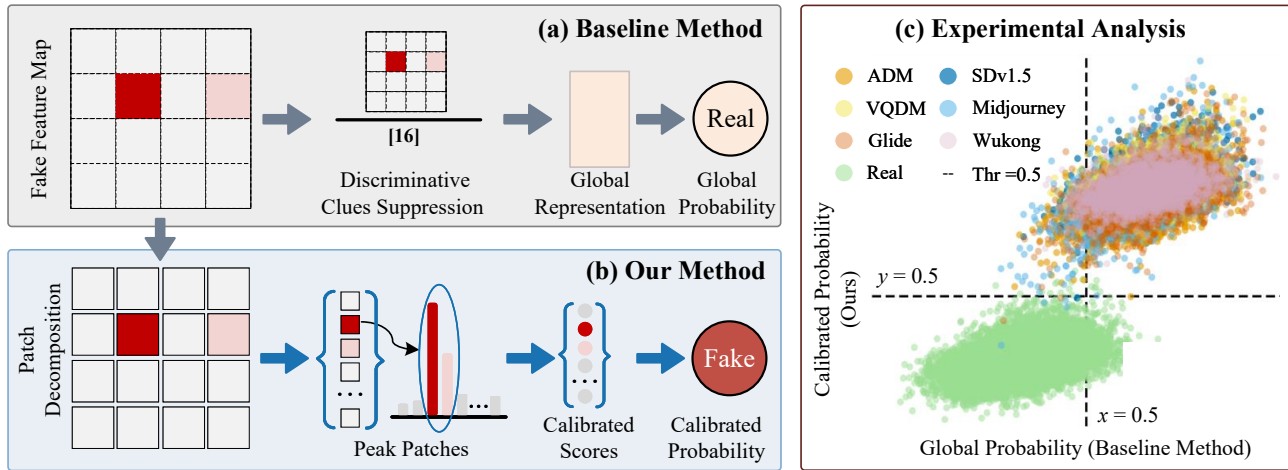

*Figure 2.* **Comparison between the conventional global representation paradigm and our proposed approach.** **(a)** In global representations, subtle discriminative artifacts (red) are suppressed by the dominant high-fidelity foreground. **(b)** Our PGC framework highlights peak feature regions to calibrate the global representation, thereby amplifying subtle discriminative traces. **(c)** As a result, high-fidelity samples initially misclassified by the baseline ($x < 0.5$) are correctly classified ($y > 0.5$).

attention shifts toward the background (e.g., non-subject areas) in AI-generated samples.[1] This indicates that the high-fidelity foreground acts as a semantic distractor, obscuring the discriminative traces required for robust detection.

To address this limitation, we propose the Peak-Guided Calibration (PGC) framework. Diverging from the reliance on global contexts, PGC is designed to capture subtle local discriminative clues. Specifically, it performs a dense evaluation across spatial patches, identifying and amplifying "peak patches" that exhibit the most critical evidence for classification. These localized peaks are then utilized to calibrate the global decision. As shown in Figure 2, this approach counters the masking effect of the realistic foreground, amplifying the subtle forgery traces required for robust detection. Specifically, while the baseline method (X-axis) misclassifies high-quality samples as "Real" due to the overwhelming realistic content, our method (Y-axis) successfully rectifies these predictions by preserving and amplifying the critical local evidence.

To better simulate real-world threats overlooked by existing datasets, we introduce the **Commercial Generation benchmark (CommGen15)**. The CommGen15 includes samples from 15 commercial models (e.g., Sora, Kling, Google Imagen), capturing the intricate post-processing and diverse degradations characteristic of "In-the-Wild" scenarios.

Our main contributions are summarized as follows:

- We propose the **Peak-Guided Calibration (PGC)** framework, which leverages local "peak patches" to calibrate the global representation, effectively cap-

---

[1]More examples are provided in Appendix Figs. 10 and 11.

turing fine-grained discriminative traces that are suppressed by high-fidelity content.

- We introduce **CommGen15**, a comprehensive benchmark collected from 15 state-of-the-art commercial models, designed to better simulate high-fidelity and complex real-world threats.

- Extensive experiments demonstrate that PGC achieves state-of-the-art performance, surpassing existing detectors by **+12.3%** on CommGen15 while demonstrating superior generalization on standard benchmarks, including GenImage (**+2.1%**), AIGI (**+3.5%**), and UniversalFakeDetect (**+1.7%**) datasets.

## 2. Related Work

### 2.1. Specialized Forensic Representations

Early detection methods focused on extracting specific fingerprints from transformation domains to distinguish generated images (Miao et al., 2022; Shi et al., 2025; Zheng et al., 2024). Approaches like FreqNet (Tan et al., 2024a) utilize Fourier or wavelet transforms to uncover spectral artifacts caused by up-sampling operations, while methods in the spatial domain (Zhao et al., 2023) exploit gradient maps (Tan et al., 2023), neighboring pixel relationships (Tan et al., 2024b), or statistical entropy (Cozzolino et al., 2024b) to capture structural inconsistencies. Recently, recognizing the limitations of holistic analysis, methods such as SAFE (Li et al., 2025a), LaDeDa (Cavia et al., 2024), and AIDE (Yan et al., 2025a) have shifted focus toward capturing micro-scale anomalies through cropping strategies or regional awareness. However, these approaches predominantly assign uniform importance to varying spatial regions.

*Table 1.* **Statistics of CommGen15.** For video subsets, the count denotes "Frames (Videos)" (e.g., 2615 frames extracted from 358 Akool videos). *Note: While Sora is primarily a video generation model, we include 1275 samples collected from community repositories (e.g., PromptHero) where they are explicitly generated or rendered as static images.*

| Type | Platform Name | Count | Original Format |
|------|--------------|-------|-----------------|
| Image | ChatGPT | 355 | PNG, JPEG |
| | Flux | 2106 | PNG, JPG |
| | Google Imagen | 1942 | PNG, JPG |
| | Ideogram | 1120 | PNG |
| | Lexica | 9525 | WEBP |
| | Midjourney | 1106 | PNG |
| | Nano Banana | 388 | JPG, PNG |
| | Sora | 1275 | PNG |
| | Stable Diffusion | 1655 | PNG, JPG |
| Video | Akool | 2615 (358) | MP4 |
| | Doubao | 2187 (729) | MP4 |
| | Hailuo | 3776 (1390) | MP4 |
| | Hunyuan | 240 (49) | MP4 |
| | Kling | 3932 (2184) | MP4 |
| | Veo | 1172 (420) | MP4 |

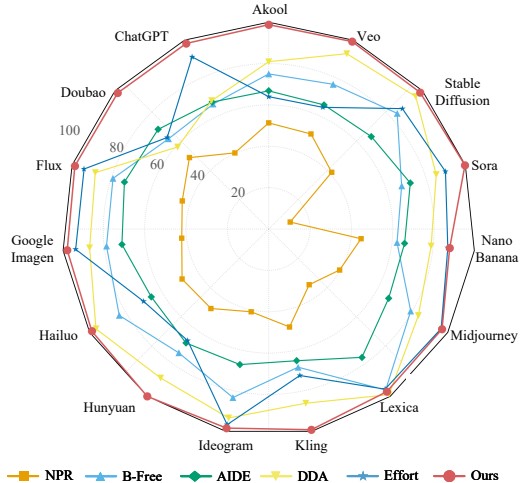

*Figure 3.* **Accuracy (%) comparison on CommGen15.** The radar chart highlights that existing methods (inner lines) struggle to generalize across diverse commercial models, whereas our method (outermost red line) maintains robust performance.

They lack a strategy to leverage discriminative "peak" features, leaving critical localized artifacts prone to being submerged by the dominant realistic content.

### 2.2. Foundation Model Adaptation

To address the rapid evolution of generative architectures, a dominant trend involves adapting pre-trained foundation models (e.g., CLIP (Radford et al., 2021)) to enhance generalization (Lu et al., 2023). Early approaches like UnivFD (Ojha et al., 2023) and ClipBased (Cozzolino et al., 2024a) demonstrated that the rich visual priors encapsulated in vision-language models possess surprising transferability across unseen generators. Building on this, recent works such as FatFormer (Liu et al., 2024), Bi-LORA (Keita et al., 2025), and Effort (Yan et al., 2025b) employ Parameter-Efficient Fine-Tuning strategies—ranging from adapters to low-rank adaptation—to further align these representations with forensic tasks. Nevertheless, such strategies retain the bias of global representations toward high-fidelity content, thereby suppressing subtle local artifacts as discussed in Section 1.

### 2.3. Data-Centric Forensics

The quality and alignment of training data have proven crucial for detector robustness. To prevent overfitting to semantic content or spurious correlations, recent studies propose constructing aligned datasets through reconstruction or conditioning mechanisms (Chen et al., 2025; Guillaro et al., 2025; Rajan et al., 2025), utilizing diffusion inversion to mine hard samples (Chen et al., 2024), or ex-

plicitly decoupling semantic-agnostic artifacts (Tao et al., 2025). Although some initiatives have targeted social media contexts (Huang et al., 2025) or continuous model evolution (Epstein et al., 2023), existing benchmarks predominantly rely on open-source generators. They fail to capture the proprietary traits of commercial platforms (e.g., unknown post-processing pipelines).

## 3. CommGen15

**Data Collection.** We introduce **CommGen15**, a benchmark comprising samples from **15 commercial models**. To capture authentic characteristics of "In-the-Wild" data, we collect samples from official public galleries and community repositories (e.g., PromptHero), avoiding local generation. This acquisition strategy ensures the preservation of platform-specific characteristics, including proprietary post-processing, web-standard compression, and invisible watermarks—factors typically absent in raw local inference. For the real image subset, we sample an equivalent number of images from the COCO 2017 dataset (Lin et al., 2014) to establish a balanced binary classification task.

**Data Processing.** We implement a unified processing pipeline to standardize the diverse raw inputs. For video models, frames are sampled every 30 frames for Hunyuan and every 60 frames for others. All images are center-cropped to $320 \times 320$ pixels and saved in lossless PNG format. This standardized protocol eliminates potential biases introduced by processing inconsistencies, such as aspect ratio variations or secondary compression noise. Table 1 shows the statistics of the dataset.[2]

---

[2]Detailed source URLs are provided in Appendix Table 10.

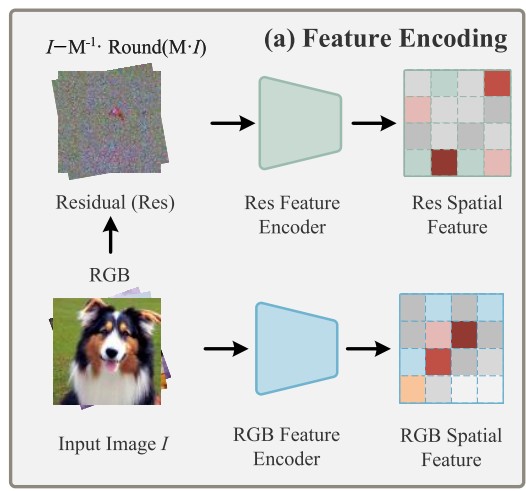 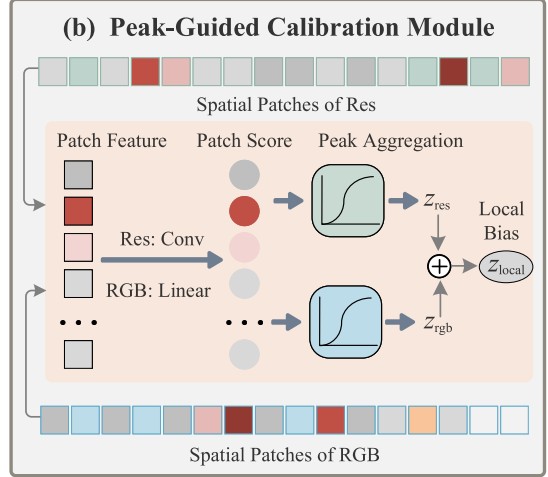 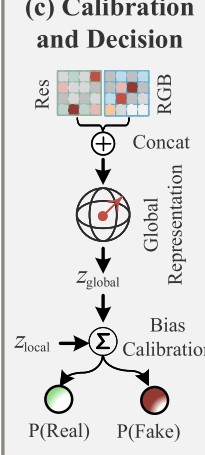

*Figure 4.* **Overview of the PGC framework.** (a) **Feature Encoding**: Extracts spatial features from residual and RGB domains via a dual-stream architecture. (b) **Peak-Guided Calibration Module**: Treats features as patch grids and aggregates the most salient artifact (peak) patches into a local bias ($Z_{\text{local}}$), ensuring decisive classification clues are not overshadowed by the dominant high-fidelity foreground content. (c) **Calibration and Decision**: This local bias additively calibrates the global logit ($Z_{\text{global}}$).

**Benchmarking Analysis.** To quantify the domain gap presented by commercial models, we evaluate five SOTA detectors on CommGen15. Specifically, AIDE, Effort, NPR, and our method are trained on GenImage (Zhu et al., 2023) (SDv1.4 (Rombach et al., 2022)), while DDA (Chen et al., 2025) and B-Free (Guillaro et al., 2025) utilize their respective synthetic datasets. As illustrated in Figure 3, existing methods (inner lines) exhibit significant performance degradation when generalizing to these unseen commercial platforms. In contrast, our method (outermost red line) maintains robust performance. This performance margin demonstrates the generalization capability of our proposed framework on unseen commercial generators.

## 4. Methodology

Global representations often fail on high-fidelity forgeries as realistic content overshadows subtle clues. We propose **Peak-Guided Calibration (PGC)** (Figure 4), which leverages discriminative local clues to calibrate global decision.

### 4.1. Feature Encoding

To detect subtle artifacts hidden in the realistic context, we employ a dual-stream architecture that extracts features from both residual and semantic domains.

**Residual Stream.** To capture low-level inconsistencies that violate the imaging physics of real cameras, we extract the quantization residual features in the RGB color space. The residual $\boldsymbol{I}_{\text{res}}$ is defined as the difference between the input image $\boldsymbol{I}$ and its re-quantized version:

$$\boldsymbol{I}_{\text{res}} = \boldsymbol{I} - \mathbf{M}^{-1} \cdot \text{Round}(\mathbf{M} \cdot \boldsymbol{I}), \qquad (1)$$

where $\mathbf{M} = [0.299, 0.587, 0.114; -0.168736, -0.331264, 0.5; 0.5, -0.418688, -0.081312]$, and $\mathbf{M}^{-1} = [1.0, 0.0, 1.402; 1.0, -0.344136, -0.714136; 1.0, 1.772, 0.0]$ denote the transformation matrix for RGB-to-YCbCr conversion and its inverse, respectively, and $\text{Round}(\cdot)$ simulates the integer quantization. This residual map $\boldsymbol{I}_{\text{res}}$ is then processed by a lightweight CNN to yield the spatial feature $\boldsymbol{F}_{\text{res}}$. Specifically, the CNN employs a stacked architecture of three convolutional blocks, each equipped with Batch Normalization and ReLU activation. **RGB Stream.** To capture high-level semantic artifacts, we utilize a pre-trained ViT (e.g., DINOv2 (Oquab et al., 2024)) as the extractor. We feed the input image through the backbone and obtain the spatial feature map $\boldsymbol{F}_{\text{rgb}}$ from the last transformer block.

### 4.2. Peak-Guided Calibration Module (PGCM)

The PGCM is designed to solve the problem where discriminative clues are submerged by the realistic context.

**Patch Partition and Scoring.** We treat feature maps $\boldsymbol{F}_{\text{res}}$ and $\boldsymbol{F}_{\text{rgb}}$ as grids of local patches. To identify anomalous regions, we project them through stream-specific heads to generate score maps $\boldsymbol{S} \in \mathbb{R}^N$:

$$\boldsymbol{S}^{\text{res}} = \Phi_{\text{Conv}}(\boldsymbol{F}_{\text{res}}), \quad \boldsymbol{S}^{\text{rgb}} = \Phi_{\text{Linear}}(\boldsymbol{F}_{\text{rgb}}), \qquad (2)$$

where $N$ is the number of patches. The score $s_i$ quantifies the forensic salience of each patch. A high score pinpoints a "peak" region rich in discriminative clues, identifying the most critical evidence for calibration, whereas a low score corresponds to areas that offer weak informative signals for forgery judgment.

**Peak Aggregation.** To prevent local classification clues from being submerged by the high-fidelity content during aggregation, we introduce a peak-focusing mechanism. This strategy enables the model to prioritize the most discriminative evidence for classification. We define the aggregated score $Z_{\text{res}}$ as:

$$Z_{\text{res}} = \tau \log\left(\frac{1}{N}\sum_{i=1}^{N}\exp(s_i^{\text{res}}/\tau)\right), \qquad (3)$$

where $\tau$ is a temperature parameter. A smaller $\tau$ sharpens the aggregation distribution, approximating the max operator. This compels the network to focus on the regions with the strongest forensic evidence ("peaks"). We apply this aggregation to both RGB ($Z_{\text{rgb}}$) and Residual ($Z_{\text{res}}$) streams and fuse them to derive the calibration bias:

$$Z_{\text{local}} = Z_{\text{res}} + \lambda \cdot Z_{\text{rgb}}, \qquad (4)$$

where $\lambda$ is a learnable scalar. Since residual artifacts are subtle, we treat $Z_{\text{res}}$ as an anchor and use $\lambda$ to modulate the stronger RGB signals, preventing high-energy semantics from overwhelming the forensic evidence.

### 4.3. Calibration and Decision

We first concatenate the spatial features $\boldsymbol{F}_{\text{res}}$ and $\boldsymbol{F}_{\text{rgb}}$ to derive a unified global feature $\boldsymbol{F}_{\text{global}}$. We then obtain a global logit $Z_{\text{global}} = \mathcal{C}(\boldsymbol{F}_{\text{global}})$ via a standard classifier head. The final prediction logit $y_{\text{pred}}$ is formulated by injecting the aggregated local bias into the global decision:

$$y_{\text{pred}} = Z_{\text{global}} + Z_{\text{local}}. \qquad (5)$$

This additive mechanism enables a decisive correction: even if the global logit $Z_{\text{global}}$ leans toward "Real" due to the predominantly realistic context, a high $Z_{\text{local}}$—indicating the presence of peak forgery artifacts—can effectively shift the decision boundary to correctly identify the forgery.

### 4.4. Optimization Objective

We optimize the PGC framework by minimizing the Binary Cross-Entropy (BCE) loss between the ground truth label $y \in \{0,1\}$ and the predicted probability $\hat{y}$:

$$\mathcal{L} = -\left[y \cdot \log(\hat{y}) + (1-y) \cdot \log(1-\hat{y})\right], \qquad (6)$$

where $\hat{y} = \sigma(y_{\text{pred}})$ represents the probability obtained by applying the sigmoid activation function $\sigma(\cdot)$ to the final predicted logit $y_{\text{pred}}$. Optimizing the fused logit $y_{\text{pred}}$ enables end-to-end training to dynamically balance the global context and local calibration. This joint supervision updates the scalar $\lambda$ and stream-specific heads, compelling the local bias $Z_{\text{local}}$ to correct global misjudgments.

## 5. Experiment

### 5.1. Experimental Setup

**Datasets. (1) Training Sets.** Following existing methods (Chen et al., 2025; Li et al., 2025b; Wang et al., 2020; Yan et al., 2025b), we utilize SDv1.4 (Rombach et al., 2022) (for Tables 2 and 3), ProGAN (Karras et al., 2018) (for Table 5), and a combination of both (for Table 4) as training data. Note that methods of B-Free (Guillaro et al., 2025) and DDA (Chen et al., 2025) are exceptions, as they require pairs of SDv2.1-generated samples and COCO images. **(2) Testing Sets.** We evaluate generalization on three standard benchmarks: **GenImage** (Zhu et al., 2023), **UniversalFakeDetect** (Ojha et al., 2023), and **AIGI** (Li et al., 2025b). We further evaluate performance on our proposed **CommGen15** dataset, a balanced real/fake benchmark that contains high-fidelity generated samples from 15 commercial models, together with an equal number of real images sampled from COCO 2017 (Lin et al., 2014). Details are provided in Section 3.

**Evaluation Baselines.** We benchmark PGC against 14 competitive detectors, categorized into: (1) Classic and Recent Baselines (2020-2023): We include widely cited methods such as CNNDet (Wang et al., 2020), FreDect (Frank et al., 2020), LGrad (Tan et al., 2023), UnivFD (Ojha et al., 2023), and PatchCraft (Zhong et al., 2023). (2) State-of-the-Art Methods (2024-2025): To demonstrate superiority over the latest techniques, we compare against FreqNet (Tan et al., 2024a), NPR (Tan et al., 2024b), FatFormer (Liu et al., 2024), AIDE (Yan et al., 2025a), CoD (Jia et al., 2025), B-Free (Guillaro et al., 2025), DDA (Chen et al., 2025), Effort (Yan et al., 2025b), and SAFE (Li et al., 2025a).

**Evaluation Metrics.** Following prior studies (Li et al., 2025a; Yan et al., 2025b), we report Average Precision (AP) to measure the precision-recall trade-off, and Accuracy (Acc) computed with a fixed threshold of 0.5.

**Implementation Details.** We employ DINOv2-Large (Oquab et al., 2024) with LoRA ($r = 8$, $\alpha = 1$) as the feature extractor. Input images are center-cropped to $224 \times 224$. The model is trained using AdamW with a batch size of 32 and a learning rate of $5 \times 10^{-5}$. For hyperparameters, we set $\tau$ to 0.5. Training takes approximately 6 hours on a single NVIDIA RTX 4090 GPU.

### 5.2. Generalization Comparisons

**Evaluation on GenImage.** We first evaluate cross-generator generalization on the GenImage benchmark (Table 2). PGC establishes a new state-of-the-art, achieving 98.3% mean Acc and 100.0% mean AP across 8 test subsets. Notably, our method surpasses the previous leading contender, CoD, by **+2.1%** in mAcc (98.3% vs. 96.2%), demonstrating superior adaptability to unseen generators.

*Table 2.* **Generalization Evaluation on GenImage (Acc and AP, %).** The detectors are trained on SDv1.4.

| Method | Ref | Midjourney | | SDv1.4 | | SDv1.5 | | ADM | | Glide | | Wukong | | VQDM | | BigGAN | | Mean | |
|---|---|---|---|---|---|---|---|---|---|---|---|---|---|---|---|---|---|---|---|
| | | Acc | AP | Acc | AP | Acc | AP | Acc | AP | Acc | AP | Acc | AP | Acc | AP | Acc | AP | Acc | AP |
| CNNDet | CVPR 2020 | 50.1 | 53.4 | 50.3 | 55.9 | 50.3 | 56.1 | 53.0 | 69.2 | 51.7 | 66.9 | 51.4 | 62.4 | 50.0 | 53.5 | 69.8 | 91.5 | 53.3 | 63.6 |
| FreDect | ICML 2020 | 32.1 | 35.7 | 28.8 | 34.9 | 28.9 | 34.6 | 62.9 | 70.1 | 42.8 | 42.2 | 35.9 | 38.0 | 72.1 | 84.2 | 26.1 | 34.7 | 41.2 | 46.8 |
| LGrad | CVPR 2023 | 75.7 | 77.5 | 76.3 | 80.1 | 77.4 | 80.1 | 51.8 | 51.0 | 49.8 | 50.5 | 73.1 | 75.4 | 52.1 | 51.5 | 40.5 | 30.2 | 61.8 | 62.0 |
| UnivFD | CVPR 2023 | 56.9 | 68.5 | 65.1 | 81.5 | 64.7 | 81.0 | 69.2 | 84.2 | 60.1 | 73.5 | 73.5 | 89.0 | 86.0 | 95.0 | 89.3 | 97.0 | 70.6 | 83.7 |
| PatchCraft | Arxiv 2023 | 89.7 | 96.2 | 95.0 | 98.9 | 94.6 | 98.8 | 81.6 | 93.3 | 83.5 | 93.8 | 90.9 | 97.4 | 88.2 | 95.9 | 91.5 | 97.8 | 89.4 | 96.5 |
| FreqNet | AAAI 2024 | 69.7 | 78.5 | 64.2 | 74.5 | 64.9 | 75.6 | 83.5 | 92.0 | 81.2 | 88.5 | 57.8 | 67.0 | 81.4 | 90.0 | 90.5 | 95.0 | 74.2 | 82.6 |
| NPR | CVPR 2024 | 77.8 | 85.4 | 78.6 | 84.0 | 78.9 | 84.6 | 69.7 | 74.6 | 78.4 | 85.7 | 76.1 | 80.5 | 78.1 | 81.0 | 80.1 | 88.2 | 77.2 | 83.0 |
| FatFormer | CVPR 2024 | 56.0 | 62.7 | 67.8 | 81.1 | 67.3 | 81.4 | 78.2 | 91.5 | 87.9 | 95.5 | 73.0 | 85.7 | 86.8 | 96.9 | 96.7 | 99.0 | 76.7 | 86.7 |
| AIDE | ICLR 2025 | 81.4 | 98.0 | 99.8 | **100.0** | 99.8 | **100.0** | 78.5 | 94.6 | 91.8 | 99.1 | 98.9 | **100.0** | 80.2 | 97.1 | 66.8 | 93.5 | 87.1 | 97.8 |
| B-Free | CVPR 2025 | 94.9 | 99.0 | 98.8 | **100.0** | 98.9 | **100.0** | 78.1 | 92.4 | 83.8 | 96.1 | 98.8 | **100.0** | 88.3 | 97.1 | 68.6 | 92.6 | 88.8 | 97.1 |
| DDA | NeurIPS 2025 | 96.3 | 99.6 | 98.0 | 99.9 | 98.1 | 99.9 | 88.4 | 97.1 | 86.5 | 96.8 | 97.7 | 99.9 | 63.1 | 81.3 | 76.3 | 92.3 | 88.1 | 95.8 |
| SAFE | KDD 2025 | 95.2 | 99.0 | 99.4 | 99.1 | 99.3 | 99.7 | 82.2 | 96.7 | 96.2 | 99.3 | 98.1 | 99.8 | 96.2 | 99.4 | 97.7 | 99.8 | 95.5 | 99.1 |
| CoD | CVPR 2025 | 96.0 | 99.0 | 99.7 | 99.9 | 99.8 | **100.0** | 85.2 | 97.4 | 95.9 | 99.2 | 98.2 | 99.9 | 96.8 | 99.9 | **98.3** | **99.9** | 96.2 | 99.4 |
| Ours | – | **100.0** | **100.0** | **100.0** | **100.0** | **100.0** | **100.0** | **100.0** | **100.0** | **100.0** | **100.0** | **100.0** | **100.0** | **100.0** | **100.0** | 86.7 | 99.7 | **98.3** | **100.0** |

*Table 3.* **Generalization Evaluation on CommGen15 (R.Acc and F.Acc, %).** The detectors are trained on SDv1.4. R.Acc and F.Acc denote real accuracy and fake accuracy, respectively.

| Method | Ref | Akool | | ChatGPT | | Doubao | | Flux | | Google Imagen | | Hailuo | | Hunyuan | | Ideogram | |
|---|---|---|---|---|---|---|---|---|---|---|---|---|---|---|---|---|---|
| | | R.Acc | F.Acc | R.Acc | F.Acc | R.Acc | F.Acc | R.Acc | F.Acc | R.Acc | F.Acc | R.Acc | F.Acc | R.Acc | F.Acc | R.Acc | F.Acc |
| NPR | CVPR 2024 | 2.7 | **99.9** | 4.2 | 76.3 | 3.3 | **99.9** | 3.4 | 84.6 | 2.8 | 81.7 | 2.5 | 94.4 | 1.7 | 93.3 | 2.8 | 79.1 |
| B-Free | CVPR 2025 | 95.1 | 54.9 | 94.1 | 38.0 | 94.5 | 35.9 | 94.8 | 63.6 | 94.6 | 63.2 | 95.0 | 72.0 | 98.3 | 49.6 | 96.0 | 70.6 |
| AIDE | ICLR 2025 | **100.0** | 33.7 | **100.0** | 34.4 | 99.9 | 44.2 | **100.0** | 46.9 | **100.0** | 42.8 | 99.9 | 31.2 | **100.0** | 36.3 | **100.0** | 34.0 |
| Effort | ICML 2025 | 98.2 | 29.8 | 98.3 | 83.9 | 97.6 | 34.6 | 98.1 | 89.7 | 97.6 | 90.2 | 98.1 | 41.7 | 98.8 | 34.6 | 97.3 | 96.1 |
| DDA | NeurIPS 2025 | 98.6 | 63.3 | 98.9 | 36.9 | 98.3 | 20.4 | 98.5 | 77.9 | 98.5 | 75.6 | 98.5 | 94.4 | 98.3 | 79.6 | 98.8 | 88.0 |
| Ours | – | **100.0** | 97.5 | **100.0** | **96.3** | **100.0** | 96.6 | 99.9 | **97.3** | **100.0** | **96.0** | **100.0** | **97.6** | **100.0** | **100.0** | **100.0** | **96.8** |

| Method | Ref | Kling | | Lexica | | Midjourney | | Nano Banana | | Sora | | Stable Diffusion | | Veo | | Mean | |
|---|---|---|---|---|---|---|---|---|---|---|---|---|---|---|---|---|---|
| | | R.Acc | F.Acc | R.Acc | F.Acc | R.Acc | F.Acc | R.Acc | F.Acc | R.Acc | F.Acc | R.Acc | F.Acc | R.Acc | F.Acc | R.Acc | F.Acc |
| NPR | CVPR 2024 | 2.5 | 94.4 | 2.4 | 64.3 | 2.5 | 76.7 | 4.1 | **85.6** | 2.5 | 19.4 | 1.8 | 80.2 | 3.1 | 97.6 | 2.8 | 81.8 |
| B-Free | CVPR 2025 | 94.7 | 42.0 | 94.6 | 97.9 | 95.1 | 63.6 | 95.9 | 28.9 | 94.5 | 40.7 | 95.1 | 72.1 | 94.7 | 58.5 | 95.1 | 56.8 |
| AIDE | ICLR 2025 | **100.0** | 30.2 | 99.9 | 53.6 | **100.0** | 33.9 | **100.0** | 32.2 | 99.9 | 44.2 | 99.9 | 33.7 | 99.8 | 31.5 | **100.0** | 37.5 |
| Effort | ICML 2025 | 98.1 | 46.8 | 98.0 | 93.4 | 98.5 | **94.1** | 96.9 | 77.3 | 97.9 | 81.9 | 97.7 | 76.4 | 98.2 | 30.6 | 97.9 | 66.7 |
| DDA | NeurIPS 2025 | 98.6 | 73.6 | 98.7 | **99.2** | 98.6 | 68.6 | 99.0 | 59.0 | 98.7 | 71.8 | 98.7 | 92.8 | 98.3 | 87.4 | 98.6 | 72.6 |
| Ours | – | **100.0** | **98.3** | **100.0** | 94.5 | **100.0** | 93.7 | **100.0** | 76.0 | 99.9 | **99.6** | **100.0** | **97.2** | **100.0** | **98.3** | **100.0** | **95.7** |

**Evaluation on CommGen15.** To assess generalization in "In-the-Wild" scenarios, we evaluate several state-of-the-art methods on our proposed CommGen15 benchmark (Table 3). The results expose a critical limitation in existing detectors: a severe performance imbalance between real and fake categories. For instance, while NPR achieves a high Fake Accuracy (F.Acc) of 81.8%, its Real Accuracy (R.Acc) collapses to a mere 2.8%, indicating a prediction bias where nearly all inputs are misclassified as fake. Conversely, methods like B-Free, AIDE, Effort, and DDA maintain high R.Acc (above 95.0%) but exhibit limited performance in detecting commercial forgeries (F.Acc less than 73%). This demonstrates that existing models struggle to generalize to the detection of high-fidelity forgeries produced by commercial models.

In contrast, PGC bridges this gap by maintaining stability on real images while achieving significant accuracy on forged ones. This balanced performance stems from our calibration mechanism, which prevents the suppression of forgery signals by high-fidelity content. Ultimately, PGC achieves a state-of-the-art overall Acc of **97.9%**, outperforming the nearest competitor (DDA) by a significant margin of **+12.3%**.[3]

**Evaluation on AIGI.** To validate scalability and cross-domain robustness, we conduct experiments on AIGI, a large-scale dataset spanning 25 diverse generative models. As detailed in Table 4, PGC demonstrates exceptional generalization, achieving a mean Acc of 86.6% and a mean AP of 95.0%. This represents a substantial improvement over the strongest baseline, DDA, surpassing it by **+3.5%** in Acc and **+5.1%** in AP.

---

[3]Detailed Acc and AP are provided in Table 12 of Appendix E.

*Table 4.* **Generalization Evaluation on AIGI (Acc and AP, %)**. The detectors are trained both on ProGAN (4 classes) and SDv1.4.

| Method | ProGAN | | R3GAN | | StyleGAN3 | | StyleGAN-XL | | StyleSwin | | WFIR | | BlendFace | | E4S | | FaceSwap | |
|---|---|---|---|---|---|---|---|---|---|---|---|---|---|---|---|---|---|---|
| | Acc | AP | Acc | AP | Acc | AP | Acc | AP | Acc | AP | Acc | AP | Acc | AP | Acc | AP | Acc | AP |
| CNNDet | 97.6 | 99.9 | 50.4 | 52.7 | 55.8 | 73.1 | 52.8 | 64.2 | 52.6 | 76.5 | 49.8 | 50.0 | 52.4 | 73.4 | 51.1 | 68.9 | 50.3 | 58.7 |
| LGrad | 96.6 | 99.8 | 54.4 | 58.7 | 70.5 | 80.5 | 65.7 | 74.6 | 81.3 | 90.0 | 51.7 | 49.4 | 41.8 | 34.9 | 41.5 | 32.8 | 45.3 | 37.5 |
| UnivFD | 98.4 | 99.9 | 83.5 | 91.2 | 79.6 | 84.5 | 84.6 | 93.3 | 86.4 | 95.2 | 70.0 | 82.0 | 35.0 | 35.3 | 57.0 | 57.1 | 53.1 | 52.4 |
| FreqNet | 99.3 | 100.0 | 62.3 | 56.8 | 83.0 | 92.4 | 79.8 | 84.1 | 80.8 | 91.8 | 58.5 | 48.9 | 23.3 | 34.1 | 25.8 | 34.7 | 40.4 | 43.4 |
| NPR | 99.4 | 100.0 | 50.8 | 61.1 | 78.4 | 91.7 | 60.3 | 75.3 | 85.7 | 94.9 | 51.6 | 65.5 | 44.5 | 34.7 | 45.0 | 34.4 | 48.1 | 43.6 |
| B-Free | 92.7 | 97.2 | 82.2 | 93.5 | 62.5 | 79.3 | 65.2 | 82.9 | 70.2 | 87.2 | 77.0 | 92.6 | 60.3 | 79.9 | 77.5 | 91.8 | 61.8 | 78.3 |
| AIDE | 97.2 | 99.6 | 92.9 | 97.1 | 88.1 | 91.4 | 88.7 | 93.2 | 83.7 | 89.3 | 71.4 | 90.8 | 51.5 | 54.2 | 44.3 | 44.3 | 52.1 | 56.3 |
| SAFE | 100.0 | 100.0 | 93.9 | 98.2 | 89.7 | 97.6 | 93.1 | 97.6 | 97.8 | 99.6 | 60.4 | 81.8 | 47.3 | 45.6 | 47.6 | 46.0 | 50.7 | 45.7 |
| DDA | 84.7 | 97.7 | 94.0 | 98.6 | 58.3 | 76.2 | 46.6 | 35.9 | 56.5 | 68.5 | 51.7 | 68.8 | 77.9 | 90.0 | 82.3 | 87.8 | 77.3 | 84.0 |
| **Ours** | 92.2 | 99.9 | 95.4 | 99.8 | 90.0 | 97.7 | 94.8 | 96.5 | 95.5 | 99.4 | 52.2 | 92.5 | 71.8 | 82.3 | 88.5 | 95.9 | 87.0 | 95.6 |

| Method | InSwap | | SimSwap | | FLUX1-dev | | Midjourney-V6 | | Glide | | DALLE-3 | | Imagen3 | | SD3 | | SDXL | |
|---|---|---|---|---|---|---|---|---|---|---|---|---|---|---|---|---|---|---|
| | Acc | AP | Acc | AP | Acc | AP | Acc | AP | Acc | AP | Acc | AP | Acc | AP | Acc | AP | Acc | AP |
| CNNDet | 54.5 | 77.9 | 52.1 | 70.0 | 57.4 | 72.3 | 52.3 | 59.8 | 51.1 | 60.0 | 53.9 | 68.6 | 51.2 | 57.4 | 55.8 | 73.1 | 52.8 | 64.2 |
| LGrad | 44.6 | 35.0 | 44.2 | 37.6 | 80.4 | 88.1 | 60.4 | 64.5 | 82.4 | 91.0 | 57.2 | 62.7 | 62.2 | 69.7 | 63.4 | 72.1 | 73.6 | 82.8 |
| UnivFD | 43.7 | 40.2 | 43.7 | 40.4 | 80.0 | 79.5 | 65.3 | 61.5 | 76.7 | 80.3 | 75.1 | 76.3 | 78.9 | 79.3 | 84.5 | 87.2 | 84.7 | 88.0 |
| FreqNet | 37.5 | 42.1 | 36.5 | 41.9 | 78.5 | 87.3 | 53.9 | 55.9 | 75.8 | 77.4 | 66.2 | 61.0 | 73.6 | 80.7 | 77.3 | 82.6 | 82.7 | 95.2 |
| NPR | 47.8 | 40.7 | 47.4 | 42.7 | 95.2 | 99.0 | 68.8 | 76.9 | 82.5 | 94.3 | 57.1 | 70.0 | 85.9 | 94.4 | 91.9 | 97.2 | 86.6 | 94.4 |
| B-Free | 62.1 | 79.6 | 63.4 | 81.2 | 53.3 | 55.0 | 60.0 | 73.3 | 83.0 | 93.1 | 78.2 | 88.6 | 50.9 | 53.2 | 69.1 | 82.8 | 89.0 | 96.3 |
| AIDE | 50.9 | 54.6 | 54.9 | 62.7 | 88.0 | 93.4 | 76.4 | 83.0 | 93.4 | 97.7 | 55.1 | 63.1 | 89.8 | 95.2 | 94.3 | 98.3 | 93.5 | 95.7 |
| SAFE | 49.7 | 49.9 | 49.0 | 49.5 | 98.1 | 99.7 | 94.1 | 98.4 | 92.5 | 97.9 | 49.0 | 45.8 | 96.7 | 98.8 | 94.1 | 98.8 | 98.3 | 99.7 |
| DDA | 82.0 | 87.8 | 83.3 | 89.8 | 95.2 | 99.3 | 96.4 | 99.5 | 85.1 | 93.7 | 93.1 | 98.1 | 90.8 | 97.2 | 95.8 | 100.0 | 95.8 | 100.0 |
| **Ours** | 80.4 | 92.3 | 82.7 | 93.5 | 81.3 | 92.2 | 85.4 | 95.8 | 96.1 | 99.7 | 86.3 | 94.9 | 86.7 | 95.6 | 96.0 | 99.6 | 96.2 | 99.9 |

| Method | BLIP | | Infinite-ID | | InstantID | | IP-Adapter | | PhotoMaker | | SocialRF | | CommunityAI | | Mean | |
|---|---|---|---|---|---|---|---|---|---|---|---|---|---|---|---|---|
| | Acc | AP | Acc | AP | Acc | AP | Acc | AP | Acc | AP | Acc | AP | Acc | AP | Acc | AP |
| CNNDet | 77.2 | 92.9 | 49.7 | 49.5 | 53.2 | 80.2 | 52.0 | 65.8 | 50.1 | 58.2 | 51.1 | 50.6 | 51.3 | 59.1 | 55.1 | 67.1 |
| LGrad | 93.0 | 97.4 | 50.9 | 54.6 | 72.6 | 81.5 | 70.3 | 78.3 | 59.9 | 67.2 | 53.5 | 54.9 | 55.5 | 69.4 | 62.9 | 66.6 |
| UnivFD | 88.6 | 95.8 | 84.5 | 89.6 | 85.4 | 93.5 | 82.6 | 87.3 | 69.3 | 72.3 | 54.4 | 55.2 | 67.0 | 73.2 | 72.5 | 75.6 |
| FreqNet | 93.8 | 100.0 | 79.0 | 74.5 | 79.8 | 86.3 | 78.8 | 79.9 | 77.0 | 74.9 | 54.2 | 58.1 | 55.9 | 69.7 | 66.1 | 70.1 |
| NPR | 99.2 | 100.0 | 63.9 | 80.4 | 63.8 | 79.2 | 82.4 | 91.7 | 48.1 | 43.6 | 59.1 | 68.4 | 54.0 | 62.9 | 67.9 | 73.5 |
| B-Free | 90.0 | 96.6 | 79.5 | 91.3 | 75.4 | 88.5 | 77.1 | 89.6 | 61.6 | 74.8 | 66.7 | 75.7 | 56.1 | 61.2 | 70.6 | 82.5 |
| AIDE | 96.4 | 95.5 | 92.2 | 94.7 | 91.8 | 96.3 | 90.0 | 95.4 | 91.7 | 95.6 | 57.8 | 65.0 | 54.1 | 61.0 | 77.6 | 82.5 |
| SAFE | 99.7 | 100.0 | 96.9 | 99.2 | 98.2 | 99.6 | 92.8 | 98.1 | 97.0 | 99.3 | 58.0 | 64.2 | 54.2 | 55.2 | 80.0 | 82.6 |
| DDA | 95.9 | 99.9 | 96.5 | 100.0 | 96.6 | 99.9 | 95.0 | 99.1 | 78.7 | 90.8 | 79.6 | 88.9 | 88.6 | 95.7 | 83.1 | 89.9 |
| **Ours** | 95.9 | 98.9 | 94.9 | 99.3 | 95.4 | 99.6 | 93.8 | 98.7 | 93.7 | 98.6 | 74.4 | 84.1 | 57.7 | 72.2 | 86.6 | 95.0 |

**Evaluation on UniversalFakeDetect.** We further extend our evaluation on the UniversalFakeDetect (Table 5). PGC sets a new record with a mean AP[4] of 98.0%, exceeding the previous best (AIDE) by **+2.2%**. Notably, while baseline methods exhibit performance degradation on specific unseen domains (e.g., AIDE on Deepfakes), our model exhibits consistent superiority across both traditional GAN architectures and modern diffusion models (e.g., LDM, Glide, DALL-E), underscoring its generalization capability.

### 5.3. Ablation Study

**Impact of Key Components.** Table 6 analyzes the contribution of each component. While the residual stream

alone serves as a strong baseline (93.4% mean Acc), the RGB stream is indispensable for specific generators like BigGAN (90.9% vs. 50.0%). However, a concatenation (Row 3) results in a performance drop to 92.1%. This empirical evidence validates our central hypothesis: global representation allows the dominant, high-fidelity content to overshadow the discriminative traces found in the residuals. By introducing our PGCM, we successfully resolve this conflict. PGCM leverages the "peak patches"—regions with the strongest trace responses—to recalibrate the global decision. This amplifies the discriminative signals, boosting the final performance to 98.3%.

**Impact of Backbone Scale.** Table 7 analyzes the scaling laws of our backbone. Scaling the model from DINOv2-Base to Large boosts the performance by 2.5%. However,

---

[4]Detailed Acc results are provided in Table 13 of Appendix E.

*Table 5.* **Generalization Evaluation on UniversalFakeDetect (AP, %).** The detectors are trained on ProGAN 4 classes.

| Method | GAN | | | | | | Deep fakes | Low level | | Perc. loss | | Guided | LDM | | | Glide | | | Dalle | Mean |
|---|---|---|---|---|---|---|---|---|---|---|---|---|---|---|---|---|---|---|---|---|
| | Pro-GAN | Cycle-GAN | Big-GAN | Style-GAN | Gau-GAN | Star-GAN | | SITD | SAN | CRN | IMLE | | 200 steps | 200 w/cfg | 100 steps | 100 27 | 50 27 | 100 10 | | |
| CNNDet | **100.0** | 93.5 | 84.5 | 99.5 | 89.5 | 98.2 | 89.0 | 73.8 | 59.5 | **98.2** | 98.4 | 73.7 | 70.6 | 71.0 | 70.5 | 80.7 | 84.9 | 82.1 | 70.6 | 83.6 |
| UnivFD | **100.0** | 98.1 | 94.5 | 86.7 | 99.3 | 99.5 | 91.7 | 78.5 | 67.5 | 83.1 | 91.1 | 79.2 | 95.8 | 79.8 | 95.9 | 93.9 | 95.1 | 94.6 | 88.5 | 90.1 |
| LGrad | **100.0** | 94.0 | 90.7 | 99.9 | 79.4 | **100.0** | 67.9 | 59.4 | 51.4 | 63.5 | 69.6 | 87.1 | 99.0 | 99.2 | 99.2 | 93.2 | 95.1 | 94.9 | 97.2 | 86.4 |
| FreqNet | 99.9 | 99.6 | 96.1 | 99.9 | 99.7 | 98.6 | 99.9 | 94.4 | 74.6 | 80.1 | 75.7 | 96.3 | 96.1 | **100.0** | 62.3 | 99.8 | 99.8 | 96.4 | 77.8 | 91.9 |
| NPR | **100.0** | 99.5 | 94.5 | 88.8 | 100.0 | 84.4 | 98.0 | 100.0 | 50.2 | 50.2 | 98.3 | 99.9 | 99.9 | 99.9 | 99.9 | 99.9 | 99.9 | 99.3 | 92.8 |
| DDA | 97.7 | 72.3 | 88.0 | 84.2 | 95.2 | 66.7 | 79.8 | 77.5 | 97.7 | 67.6 | 91.4 | 96.8 | 96.0 | 97.3 | 95.6 | 86.6 | 84.7 | 92.4 | 64.9 | 85.9 |
| AIDE | **100.0** | **99.9** | 94.4 | **100.0** | 97.7 | **100.0** | 76.2 | 77.9 | 90.3 | 91.2 | **100.0** | 97.6 | 99.3 | 99.1 | 99.3 | 99.3 | 99.3 | 99.2 | 99.0 | 95.8 |
| B-Free | 98.6 | 86.9 | 97.7 | 92.9 | 99.2 | 88.8 | 83.0 | **99.9** | 99.2 | 96.5 | 93.2 | 94.2 | 99.5 | 99.7 | 99.5 | 92.2 | 92.5 | 94.2 | 98.0 | 95.0 |
| **Ours** | 99.8 | **99.9** | **99.7** | 94.8 | **100.0** | 97.6 | 94.3 | 84.9 | 99.0 | 96.5 | 99.3 | **99.6** | 98.7 | 99.9 | 99.8 | 99.6 | 99.6 | 99.7 | **99.7** | **98.0** |

*Table 6.* **Ablation study of key components on GenImage (Acc, %).** "PGCM" denotes our Peak-Guided Calibration Module.

| Res | RGB | PGCM | Midjourney | SDv1.4 | SDv1.5 | ADM | Glide | Wukong | VQDM | BigGAN | Mean |
|---|---|---|---|---|---|---|---|---|---|---|---|
| ✓ | × | × | 98.8 | 99.9 | 99.8 | 99.5 | 99.6 | 99.7 | 100.0 | 50.0 | 93.4 |
| × | ✓ | × | 83.3 | 99.6 | 99.4 | 69.6 | 95.3 | 96.8 | 85.7 | 90.9 | 90.1 |
| ✓ | ✓ | × | 96.5 | 96.7 | 96.6 | 96.7 | 96.8 | 96.5 | 96.9 | 60.5 | 92.1 |
| ✓ | ✓ | ✓ | 100.0 | 100.0 | 100.0 | 100.0 | 100.0 | 100.0 | 100.0 | 86.7 | **98.3** |

*Table 7.* **Impact of backbone scale on GenImage (Acc, %).** Scaling from Base to Large yields significant gains, while further scaling to Giant does not improve and slightly degrades performance, validating our choice of DINOv2-Large.

| Model | Midjourney | SDv1.4 | SDv1.5 | ADM | Glide | Wukong | VQDM | BigGAN | Mean |
|---|---|---|---|---|---|---|---|---|---|
| DINOv2-Base | 99.8 | 100.0 | 99.9 | 99.9 | 99.9 | 100.0 | 100.0 | 66.6 | 95.8 |
| DINOv2-Large | 100.0 | 100.0 | 100.0 | 100.0 | 100.0 | 100.0 | 100.0 | 86.7 | **98.3** |
| DINOv2-Giant | 99.8 | 100.0 | 99.9 | 99.8 | 99.8 | 100.0 | 100.0 | 84.0 | 97.9 |

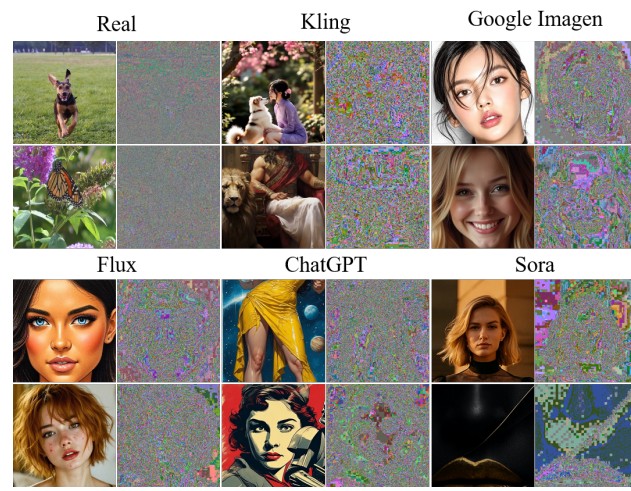

Real     Kling     Google Imagen

Flux     ChatGPT     Sora

*Figure 5.* **Visualization of residual noise artifacts.** While real images exhibit uniform, natural noise distributions, samples from commercial generators reveal distinct, localized artifact patterns in the residual domain.

further scaling to the Giant variant yields diminishing returns (97.9%). Consequently, we select DINOv2-Large to achieve an optimal trade-off between computational efficiency and detection performance.

### 5.4. Model Attribution and Robustness Analysis

**Visualization of residual noise.** We visualize the residual maps of real and generated images (Figure 5). Real images exhibit uniform noise distributions. In contrast, generated images exhibit distinct local artifact-noise patterns in the residual domain. This observation aligns with our "peak patch" assumption, confirming that forgery artifacts are not globally uniform but concentrated in specific regions.

**Foreground-Background Attribution.** As visualized in Figure 1, the detector's focus on AI-generated images is predominantly concentrated on background regions. To quantify this observation, we calculate the SHAP values for the "fake" logit and segment the images into foreground (FG) and background (BG) using SAM. To account for varying region sizes, we compute the area-normalized per-pixel average of the positive ($d^+$) and the magnitude of negative ($|d^-|$) SHAP values for each region. Here, positive values indicate evidence supporting the "fake" prediction, whereas negative values act as counter-evidence. Table 8 reports two metrics: the positive evidence gap (Pos. Gap) between the background and foreground ($d_{\text{BG}}^+ - d_{\text{FG}}^+$), and the net evidence (Net Evid.) within the foreground ($d_{\text{FG}}^+ - |d_{\text{FG}}^-|$). Across seven representative generators, the positive gap remains consistently greater than zero (mean: 0.307), while the foreground net evidence is consistently negative (mean: $-0.033$). This confirms that backgrounds provide the primary supportive evidence for fake detection, whereas foregrounds contribute minimally to the decision and often serve as counter-evidence.

**Robustness Analysis.** We evaluate our method's robustness against baselines (DDA, Effort) under five types of perturbations. These include gaussian blur ($\sigma = 2.0$), adjustments to brightness/contrast (factor of 1.15) and saturation (1.30), and shot noise, where we set the photon count

*Table 8.* **Signed SHAP evidence on foreground (FG) and background (BG) regions.** Here, $d^+$ and $d^-$ denote the per-pixel average of positive and negative SHAP values, respectively. The table reports the positive evidence gap ($d^+_{\text{BG}} - d^+_{\text{FG}}$) and the foreground net evidence ($d^+_{\text{FG}} - |d^-_{\text{FG}}|$).

| Metric | Doubao | Akool | Flux | Google Imagen | Hailuo | Nano Banana | Stable Diffusion | Mean |
|---|---|---|---|---|---|---|---|---|
| Pos. Gap | 0.244 | 0.244 | 0.275 | 0.409 | 0.273 | 0.290 | 0.411 | 0.307 |
| Net Evid. | -0.045 | -0.027 | -0.033 | -0.034 | -0.015 | -0.039 | -0.039 | -0.033 |

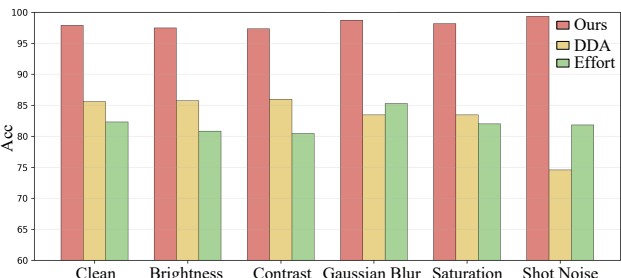

*Figure 6.* **Robustness comparison under degradations.**

scale to $p = 15$ to simulate severe low-light granularity.[5] As shown in Figure 6, while baselines like DDA suffered significant performance drops—falling below 75% Acc under shot noise—our method remained remarkably stable, consistently maintaining over 97% Acc. This demonstrates its superior robustness and feature extraction capabilities.

**Adversarial Robustness.** Beyond common image degradations, we further evaluate the robustness of PGC against transfer-based black-box adversarial attacks on CommGen15. We adopt Effort (Yan et al., 2025b), a strong state-of-the-art detector trained on the same SDv1.4 dataset, as the surrogate model. Specifically, adversarial examples are generated on Effort using 40-step PGD and then directly transferred to attack our detector, with perturbation budgets $\epsilon \in \{0.3, 0.4, 0.5\}$. As shown in Table 9, PGC maintains 98.1% mAcc under $\epsilon = 0.3$, which is nearly unchanged compared with the clean setting. Even when $\epsilon$ increases to 0.4 and 0.5, PGC still achieves 95.0% and 89.5% mAcc, respectively. These results demonstrate that our method is robust to strong transfer-based black-box PGD attacks.

## 6. Conclusion

In this work, we address the suppression of subtle discriminative traces in high-fidelity content by proposing the PGC framework. Specifically, PGC leverages local "peak patches"—regions containing critical evidence for classification—and utilizes them to recalibrate the global decision. To better simulate real-world threats, we introduce CommGen15, a benchmark from 15 commercial models. PGC

---

[5]Detailed configurations are in Appendix E.3.

*Table 9.* **Transfer-based black-box adversarial robustness on CommGen15 under 40-step PGD attacks.** Effort is used as the surrogate model. We report Acc (%).

| Epsilon ($\epsilon$) | Akool | ChatGPT | Doubao | Flux | Google Imagen | Hailuo | Hunyuan | Ideogram |
|---|---|---|---|---|---|---|---|---|
| 0 | 98.0 | 99.5 | 99.5 | 98.5 | 97.5 | 99.5 | 100.0 | 98.5 |
| 0.3 | 99.0 | 98.5 | 98.5 | 98.5 | 98.5 | 98.5 | 99.0 | 96.5 |
| 0.4 | 95.0 | 95.5 | 96.0 | 95.5 | 94.0 | 96.0 | 96.0 | 94.0 |
| 0.5 | 89.0 | 90.0 | 86.5 | 92.0 | 90.5 | 91.5 | 90.5 | 86.5 |

| Epsilon ($\epsilon$) | King | Lexica | Midjourney | Nano Banana | Sora | Stable Diffusion | Veo | Mean |
|---|---|---|---|---|---|---|---|---|
| 0 | 100.0 | 97.0 | 96.5 | 89.5 | 100.0 | 99.0 | 99.5 | 98.2 |
| 0.3 | 99.0 | 97.5 | 98.0 | 98.0 | 97.5 | 96.0 | 98.5 | 98.1 |
| 0.4 | 95.5 | 93.0 | 95.5 | 97.0 | 94.0 | 93.0 | 94.5 | 95.0 |
| 0.5 | 89.0 | 88.5 | 91.5 | 89.0 | 88.5 | 89.0 | 90.5 | 89.5 |

achieves state-of-the-art performance on standard datasets and a +12.3% gain on CommGen15, highlighting the imperative of shifting from global representation learning to peak-sensitive artifact mining.

## Impact Statement

We introduce the PGC framework and the CommGen15 benchmark to detect high-fidelity AI-generated images and video frames, combating the spread of deceptive visual content. While critical for maintaining digital trust, we acknowledge that our insights into "peak patches" could theoretically be exploited to engineer evasion attacks. To mitigate this risk, we release CommGen15 as a defensive benchmark under a research-only license, ensuring reproducibility while discouraging and explicitly prohibiting malicious use.

## Acknowledgements

This work is supported in part by the Guangdong S&T Program (2026B0101100003), the National Natural Science Foundation of China (U23B2023 and 62472199), the Guangdong Key Laboratory of Data Security and Privacy Preserving (2023B1212060036), the Guangdong-Hong Kong Joint Laboratory for Data Security and Privacy Protection (2023B1212120007), the Basic and Applied Basic Research Foundation of Guangdong Province (2025A1515011097), and the Outstanding Youth Project of the Guangdong Basic and Applied Basic Research Foundation (2023B1515020064). This work is also supported by the Engineering Research Center of Trustworthy AI, Ministry of Education.

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

# A. Appendix Organization

This appendix provides supplementary technical details and comprehensive experimental analyses to support the main paper. The content is structured as follows: **Section B** elaborates on the detailed architecture of the Peak-Guided Calibration Module (PGCM). **Section C** describes the construction of the proposed CommGen15 benchmark, including data sources and preprocessing pipelines. **Section D** outlines the experimental settings and training datasets. **Section E** presents more experimental results, covering expanded generalization metrics, hyperparameter sensitivity analysis, robustness analysis under perturbations, and qualitative visualizations. Finally, **Section F** discusses the limitations and future research directions.

# B. Detailed Architecture of the Peak-Guided Calibration Module (PGCM)

The PGCM is designed to capture localized forgery artifacts by treating the input image as a grid of local instances. A critical design characteristic of our framework is the distinct spatial partitioning strategies employed by the RGB and Residual streams, tailored to their respective feature extraction backbones.

In the RGB stream, the spatial partitioning is determined by the tokenization process of the Vision Transformer (DINOv2). The input image $\mathbf{I} \in \mathbb{R}^{H \times W \times 3}$ is partitioned into a sequence of non-overlapping patches. In our standard implementation, we adopt a patch size of $p \times p$ with $p = 14$. Consequently, for an input resolution of $224 \times 224$, the image is partitioned into a grid of $16 \times 16$, yielding a total of $N = 256$ patch tokens. These tokens are subsequently projected to a scalar anomaly score vector $\boldsymbol{S}^{\mathrm{rgb}} \in \mathbb{R}^N$, where each element represents the forgery likelihood of a specific $14 \times 14$ pixel region. This explicit partitioning allows the model to leverage the global semantic attention of ViT while maintaining local sensitivity.

Conversely, the residual stream is implemented as a fully convolutional encoder that produces a dense spatial score map. Unlike ViT tokenization with explicit non-overlapping patches, the residual stream defines its local instances by the network's cumulative downsampling stride. Specifically, the residual encoder stacks three $3 \times 3$ convolutional blocks, each with stride 2 and padding 1, resulting in an overall output stride of $2^3 = 8$. Therefore, for an input of $224 \times 224$, the feature map resolution becomes $112 \times 112 \to 56 \times 56 \to 28 \times 28$ (i.e., $H/8 \times W/8$ when (H,W) are multiples of 8). Each spatial location on this map corresponds to an input sampling step ("effective stride") of 8 pixels. Importantly, the receptive field of each feature location is larger than $8 \times 8$: with three $3 \times 3$ strided convolutions, the receptive field size is approximately $15 \times 15$

pixels in the input (BatchNorm/ReLU do not change it). A subsequent $1 \times 1$ convolutional prediction head collapses the channel dimension and produces the residual anomaly score map $\mathbf{S}^{\mathrm{res}} \in \mathbb{R}^{H/8 \times W/8}$, where each score evaluates local residual artifacts within its receptive field.

To unify these multi-granularity local evaluations into a cohesive decision, we devise a soft peak-focusing mechanism. Let $\mathcal{S} = \{s_i\}_{i=1}^N$ denote the set of local anomaly scores extracted from either stream, where $s_i$ represents the score of the $i$-th instance (patch or feature point) and $N$ is the total number of instances. The aggregated global score $Z$ is computed as Eq. (3). This formulation acts as an approximation of the max operator. Crucially, it ensures that the gradient flow is dominated by the patches with the highest artifact scores (the "peaks"), guiding the model to focus on the most discriminative regions. This aggregation strategy effectively decouples the global decision from the specific spatial resolution of the feature maps, enabling the flexible integration of the $14 \times 14$ based semantic tokens and the $8 \times 8$ based noise patterns.

# C. Details of CommGen15

To address the limitations of existing benchmarks in capturing the complexity of modern proprietary generators, we introduce the **CommGen15** benchmark. Figure 7 presents representative examples.

**Dataset Composition.** CommGen15 spans two modalities—*images* and *videos*—generated by 15 diverse commercial models (9 for images and 6 for videos). The raw data formats include `PNG`, `JPG`, `JPEG`, `WEBP`, and `MP4`. In total, the dataset comprises **33394** generated samples, consisting of 19472 synthetic images and 13922 extracted video frames. To construct a balanced benchmark, we sourced an equivalent number of **33394** authentic images from the MS COCO dataset, resulting in a grand total of **66788** samples. Detailed statistics are summarized in Table 10.

**Standardization Pipeline.** To mitigate confounding factors arising from heterogeneous resolutions and compression artifacts, we implement a unified preprocessing pipeline. For video platforms, we apply a fixed-interval frame sampling strategy to ensure temporal diversity: *Hunyuan* is sampled at a rate of 1 frame per 30 frames, while all other platforms are sampled at 1 frame per 60 frames. All raw images and extracted frames are center-cropped to $320 \times 320$ pixels and stored in lossless `PNG` format. This standardization compels detectors to focus on intrinsic generative artifacts rather than superficial cues such as aspect ratios or codec signatures.

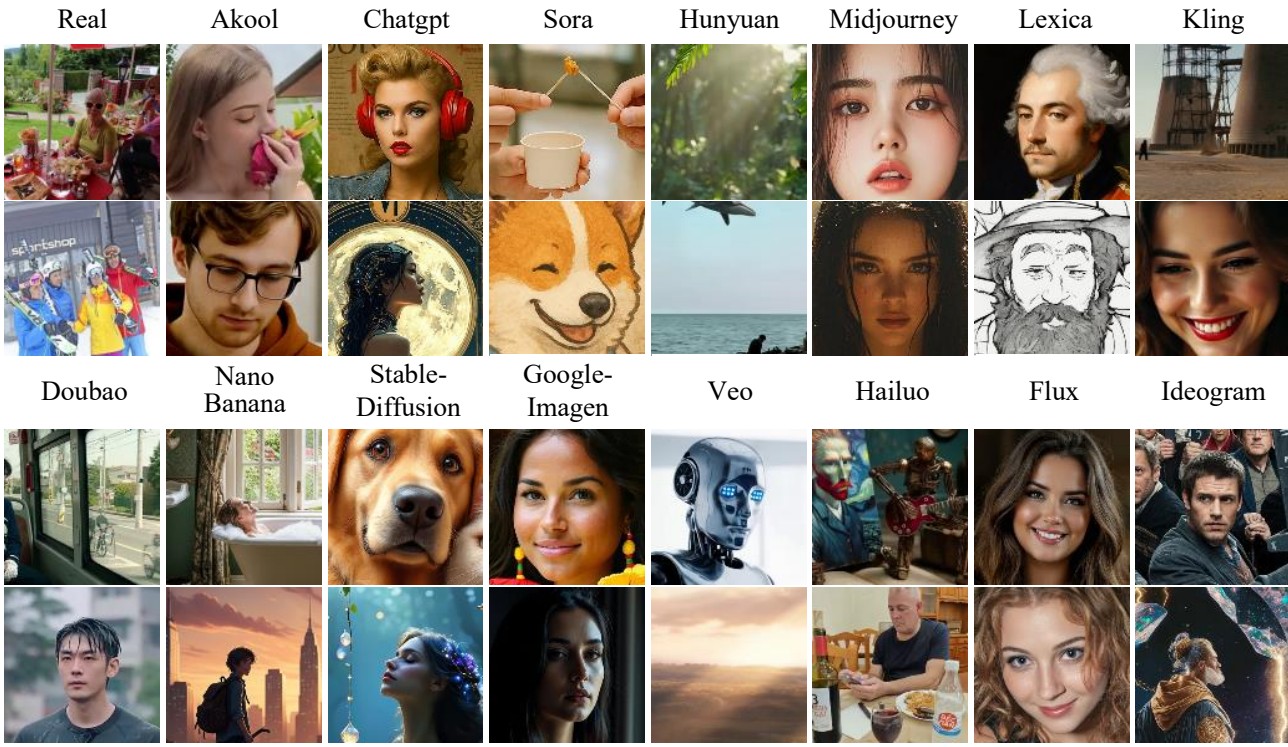

*Figure 7.* **Illustration of visual examples in CommGen15.** The dataset includes high-fidelity images and video frames from 15 leading commercial platforms, exhibiting diverse semantic content and visual styles.

### C.1. Image Platforms

**ChatGPT (GPT-4o; OpenAI).** Image generation and editing in ChatGPT are powered by *GPT-4o*, supporting text-to-image synthesis, iterative refinement conditioned on conversational context, and image-based editing (including localized modifications and inpainting). Outputs are provided in PNG/JPEG, with typical resolutions of 1024 × 1024, 1792 × 1024, and 1024 × 1792.

**Flux (FLUX.1 / FLUX.1.1; Black Forest Labs).** Flux corresponds to the FLUX model family, including *FLUX.1 [Pro/Dev/Schnell]* and *FLUX.1.1 [Pro/Ultra]*. The primary interface is text prompting, while certain versions/APIs additionally support image-conditioned controls (e.g., image-to-image, inpainting/outpainting, and multi-reference constraints). Outputs are in PNG/JPG, with common resolutions spanning diverse aspect ratios and pixel scales such as 1024 × 1024, 1344 × 768, and 2048 × 2048.

**Google Imagen (Imagen 4; Google DeepMind).** Google Imagen employs *Imagen 4* for text-to-image generation, covering both photorealistic and artistic styles. Outputs are in PNG/JPG, with resolutions up to the 2K scale.

**Ideogram (Ideogram-v3-quality / Ideogram v2 Turbo / Ideogram v2).** Ideogram is a text-to-image platform specialized for high-fidelity text rendering and layout-centric visual generation, making it suitable for design scenarios with explicit textual and compositional constraints. CommGen15 includes *ideogram-v3-quality*, *Ideogram v2 Turbo*, and *Ideogram v2*. Outputs are primarily PNG, with typical resolutions of 736 × 1312, 1104 × 1968, and 1024 × 1024.

**Lexica (Aperture Max).** Lexica is a Stable Diffusion ecosystem platform combining generation and retrieval, supporting prompt-based synthesis (including negative prompts) and a searchable gallery to encourage stylistic diversity. The CommGen15 Lexica subset is generated by *Aperture Max*. Outputs are predominantly WEBP at 1664 × 2496.

**Midjourney.** Midjourney is a text-to-image platform that supports parameterized control over aspect ratio and style strength, and optionally leverages reference images in certain workflows. Outputs are PNG; typical base generations are around 1024 × 1024, with support for alternative aspect ratios and higher-resolution upscales (e.g., 2048 × 2048 or above depending on mode and configuration).

**Nano Banana (Nano Banana (Pro); Google Gemini).** Nano Banana is an image generation and editing model family in the Gemini ecosystem, comprising *Nano Banana* and *Nano Banana Pro*. It supports instruction-driven synthesis and context-aware editing conditioned on input images. Outputs are provided in PNG/JPG, with typical resolutions

*Table 10.* **Detailed Statistics of CommGen15.** We collect data from 15 commercial models. **Count** denotes the number of images or extracted video frames (with source video count in parentheses). URLs are available as of May 16, 2026, except for Sora, whose web and app services were discontinued on April 26, 2026.

| Type | Platform Name | Count | Original Format | Platform Link | Data Source |
|---|---|---|---|---|---|
| Image | ChatGPT | 355 | PNG, JPEG | https://chatgpt.com/images | https://prompthero.com/chatgpt-image-prompts |
| | Flux | 2106 | PNG, JPG | https://bfl.ai/models | https://prompthero.com/flux-prompts |
| | Google Imagen | 1942 | PNG, JPG | https://deepmind.google/models/imagen/ | https://prompthero.com/search?model=Google+Imagen |
| | Ideogram | 1120 | PNG | https://ideogram.ai/ | https://prompthero.com/Ideogram-prompts |
| | Lexica | 9525 | WEBP | https://lexica.art/ | https://lexica.art/ |
| | Midjourney | 1106 | PNG | https://www.midjourney.com/ | https://prompthero.com/midjourney-prompts |
| | Nano Banana | 388 | JPG, PNG | https://gemini.google.com/app | https://prompthero.com/nano-banana-prompts |
| | Sora | 1275 | PNG | – | https://prompthero.com/sora-prompts |
| | Stable Diffusion | 1655 | PNG, JPG | https://stability.ai/stable-image | https://prompthero.com/stable-diffusion-prompts |
| Video | Akool | 2615 (358) | MP4 | https://akool.com/zh-cn | https://akool.com/zh-cn/community |
| | Doubao | 2187 (729) | MP4 | https://www.doubao.com/chat/ | https://www.volcengine.com/ |
| | Hailuo | 3776 (1390) | MP4 | https://hailuoai.com/ | https://prompthero.com/hailuo-prompts |
| | Hunyuan | 240 (49) | MP4 | https://hunyuan.tencent.com/ | https://prompthero.com/hunyuan-prompts |
| | Kling | 3932 (2184) | MP4 | https://klingai.com/ | https://prompthero.com/kling-prompts |
| | Veo | 1172 (420) | MP4 | https://deepmind.google/models/veo/ | https://prompthero.com/veo-prompts |

ranging from $1024 \times 1024$ to $4096 \times 4096$, including commonly used settings such as $800 \times 1280$ and $2048 \times 2048$.

**Sora (Sora v1; OpenAI).** Sora is primarily a video generation system, and also supports text-to-image and reference-conditioned image generation; the image samples in CommGen15 mainly correspond to *Sora v1*. Outputs are in PNG, with a typical resolution of $1024 \times 1536$.

**Stable Diffusion (Stable Diffusion 1.5; Stability AI).** Stable Diffusion is an open-source diffusion model family; the CommGen15 image subset uses *Stable Diffusion 1.5*. Outputs are in JPG/PNG, with a canonical base resolution of $512 \times 512$, and additional non-square outputs such as $512 \times 768$, $512 \times 760$, and $688 \times 1032$.

## C.2. Video Platforms

**Akool.** Akool is a generative content platform supporting image-to-video generation and identity-centric editing capabilities (e.g., face swapping and digital-human workflows). Outputs are MP4, with typical resolutions including 720p and $1920 \times 1080$, and optional exports up to 4K depending on configuration.

**Doubao (Volcengine).** Doubao-related video functionality is accessed via the Volcengine interface, using the Doubao-Seedance-1.0-pro model. Outputs are MP4, and the resolution varies with API configuration, commonly exported in high-definition profiles on the platform side.

**Hailuo (MiniMax).** Hailuo supports text-to-video generation as well as reference-conditioned image-to-video synthesis. Outputs are MP4, typically at 720p with multiple aspect ratios, including $720 \times 1280$, $720 \times 944$, $720 \times 720$, and $720 \times 1248$.

**Hunyuan (Tencent).** Hunyuan supports multimodal generation, including video synthesis. Outputs are MP4, with resolutions such as $512 \times 512$, $704 \times 1280$, and $960 \times 1280$, depending on the model and mode configuration.

*Table 11.* **Summary of existing datasets used in our experiments.**

| Bench. | Models | Quantity | Resolution |
|---|---|---|---|
| GenImage | Midjourney (Midjourney Team) | 12000 | 512×512 |
| | SDv1.4 (Rombach et al., 2022) | 12000 | 512×512 |
| | SDv1.5 (Rombach et al., 2022) | 16000 | 512×512 |
| | ADM (Dhariwal & Nichol, 2021) | 12000 | 256×256 |
| | Glide (Nichol et al., 2022) | 12000 | 256×256 |
| | Wukong (MindSpore, 2022) | 12000 | 512×512 |
| | VQDM (Gu et al., 2022) | 12000 | 256×256 |
| | BigGAN (Brock et al., 2018) | 12000 | 128×128 |
| AIGI | ProGAN (Karras et al., 2018) | 8000 | 256×256 |
| | R3GAN (Huang et al., 2024) | 9000 | 256×256 |
| | StyleGAN3 (Karras et al., 2021) | 9000 | 512-1024 |
| | StyleGAN-XL (Sauer et al., 2022) | 9000 | 1024×1024 |
| | StyleSwin (Zhang et al., 2022) | 9000 | 1024×1024 |
| | WFIR (West & Bergstrom, 2019) | 2000 | 1024×1024 |
| | BlendFace (Shiohara et al., 2023) | 9000 | 256×256 |
| | E4S (Liu et al., 2023) | 9000 | 256×256 |
| | FaceSwap (Marek, 2020) | 9000 | 256×256 |
| | InSwap (Wang et al., 2023) | 8900 | 256×256 |
| | SimSwap (Chen et al., 2020) | 9000 | 256×256 |
| | FLUX1-dev (Black Forest Labs, 2024) | 9000 | 1024×1024 |
| | Midjourney-V6 (Midjourney Team) | 6000 | 2048×2048 |
| | Glide (Nichol et al., 2022) | 9000 | 256×256 |
| | DALLE-3 (OpenAI Team, 2024) | 8000 | 1024×1024 |
| | Imagen3 (Google DeepMind, 2024) | 9000 | 1024×1024 |
| | SD3 (Esser et al., 2024) | 9000 | 1024×1024 |
| | SDXL (Podell et al., 2024) | 9000 | 1024×1024 |
| | BLIP (Li et al., 2022) | 9000 | 512×512 |
| | Infinite_ID (Wu et al., 2024) | 9000 | 1024×1024 |
| | InstantID (Wang et al., 2024) | 9000 | 1280×1024 |
| | IP_Adapter (Ye et al., 2023) | 9000 | 1024×1024 |
| | PhotoMaker (Li et al., 2024) | 9000 | 1024×1024 |
| | SocialRF (Li et al., 2025b) | 6000 | 448×832 - 3840×2160 |
| | CommunityAI (Li et al., 2025b) | 12000 | 450×656 - 1536×2688 |
| UniversalFakeDetect | ProGAN (Karras et al., 2018) | 8000 | 256×256 |
| | CycleGAN (Zhu et al., 2017) | 2642 | 256×256 |
| | BigGAN (Brock et al., 2018) | 4000 | 256×256 |
| | StyleGAN (Karras et al., 2019) | 11982 | 256×256 - 512×384 |
| | GauGAN (Park et al., 2019) | 10000 | 256×256 |
| | StarGAN (Choi et al., 2018) | 3998 | 256×256 |
| | Deepfakes (Rossler et al., 2019) | 5405 | 256×256 - 434×438 |
| | SITD (Chen et al., 2018) | 360 | 4256×2848 - 6030×4032 |
| | SAN (Dai et al., 2019) | 438 | 228 × 256 1280×1200 |
| | CRN (Chen & Koltun, 2017) | 12764 | 512×256 |
| | IMLE (Li et al., 2019) | 12764 | 512×256 |
| | Guided (Nichol et al., 2022) | 2000 | 256×256 |
| | LDM (200 steps) (Rombach et al., 2022) | 2000 | 256×256 |
| | LDM (200 steps w/cfg) (Rombach et al., 2022) | 2000 | 256×256 |
| | LDM (100 steps) (Rombach et al., 2022) | 2000 | 256×256 |
| | Glide (100-27) (Nichol et al., 2022) | 2000 | 256×256 |
| | Glide (50-27) (Nichol et al., 2022) | 2000 | 256×256 |
| | Glide (100-10) (Nichol et al., 2022) | 2000 | 256×256 |
| | Dalle (Ramesh et al., 2021) | 2000 | 256×256 |

**Kling (v1.6).** Kling AI (v1.6) supports multimodally conditioned video generation and enhancement. Outputs are MP4, with typical resolutions including $1088 \times 1888$, $1440 \times 1440$, and $1920 \times 1080$.

**Veo (Veo-2 / Veo-3; Google DeepMind).** The Veo series supports text- and reference-conditioned video generation. Outputs are MP4, with common resolutions including $1280 \times 720$ and $1920 \times 1080$.

# D. Experimental Setup and Details

## D.1. Datasets

**Training Datasets.** For the detector trained on images generated by ProGAN (Karras et al., 2018), we use the training split released by ForenSynths (Wang et al., 2020). The dataset includes ProGAN-generated images from four categories, namely *car, cat, chair*, and *horse*, together with real images sampled from LSUN (Yu et al., 2015). In total, 144,000 images are used for training. For the detector trained on images generated by SDv1.4, we adopt the training set from GenImage (Zhu et al., 2023). This dataset contains 324,000 images with a resolution of $512 \times 512$, where the real images are sourced from ImageNet (Deng et al., 2009).

**Evaluation Benchmarks.** We evaluate generalization across three established benchmarks—**GenImage** (Zhu et al., 2023), **AIGI** (Li et al., 2025b), and **UniversalFakeDetect** (Ojha et al., 2023)—in addition to our proposed **CommGen15**. Table 11 summarizes the statistics of the existing benchmarks used in this paper.

# E. More Experimental Results

## E.1. Expanded Generalization Analysis

**CommGen15 (Acc and AP).** Table 12 details the Accuracy (Acc) and Average Precision (AP) on CommGen15. While the main text discusses the trade-off between Real Accuracy (R.Acc) and Fake Accuracy (F.Acc), this table provides a holistic view. As shown in Table 12, our method achieves 97.9% mean Acc and 99.0% mean AP across 15 commercial generators, establishing the best overall performance. Compared with the strongest baseline DDA (85.6% mean Acc, 97.0% mean AP), our approach improves mean Acc by **12.3%** and increases mean AP by **2.0%**.

**UniversalFakeDetect (Acc).** Table 13 supplements the main text by reporting Acc on UniversalFakeDetect. PGC achieves the highest mean Acc of 90.6%, outperforming the strongest baseline B-Free (88.9%) by **+1.7%**. This reinforces our method's generalizability across both GAN-based and Diffusion-based architectures.

## E.2. Hyperparameter Sensitivity Analysis

We investigate the impact of key hyperparameters to validate our design choices. Specifically, we analyze the patch resolution for the RGB stream and the smoothing temperature $\tau$ for the peak aggregation.

**Impact of Patch Resolution ($p$).** To determine the optimal spatial configuration, we varied the patch size $p$ from 14 to 112. Given the input resolution of $224 \times 224$, a smaller patch size yields a denser feature grid (e.g., $p = 14$ results in $16 \times 16$ tokens). As shown in Table 14, there is a clear inverse correlation between patch size and detection accuracy. The standard $14 \times 14$ setting achieves the best performance (98.3%), while increasing $p$ to 112 degrades accuracy to 92.1%. This confirms that a fine-grained grid is essential for capturing localized forgery traces, validating our choice of the $14 \times 14$ configuration.

**Impact of Smoothing Parameter ($\tau$).** The temperature $\tau$ in Eq. 3 controls the sharpness of the peak aggregation. Table 15 reports the results across different $\tau$ values. We observe that setting $\tau = 0.25$ approximates the max operator too aggressively, making the model sensitive to local noise (95.1% Acc). Conversely, a large $\tau = 1.00$ overly smooths the aggregation, causing subtle artifacts to be submerged by the background (96.4% Acc). The model achieves optimal performance at $\tau = 0.5$ (98.3% Acc), striking the best balance between highlighting salient anomalies and maintaining gradients.

## E.3. Robustness Analysis

To assess reliability in uncontrolled environments, we evaluate the detector (trained on SDv1.4) on CommGen15 under 12 distinct perturbations. These perturbations simulate common degradations encountered in social media transmission and post-processing. We test five severity levels for each perturbation type. Figure 8 visualizes these effects, and Figure 9 reports the performance curves. In the figure, "Clean" denotes the original, unperturbed input. The definitions and parameter settings of each perturbation are detailed: **Brightness & Contrast & Gamma:** Scaling factors $[0.95, 0.90, 1.05, 1.10, 1.15]$. Simulates exposure variations and nonlinear tone mapping. **Blur (Defocus & gaussian):** Radius/Sigma $[0.6 \dots 2.2]$ and $[0.4 \dots 2.0]$. Simulates lens misfocus or compression smoothing. **Motion Blur:** Kernel lengths $[3 \dots 11]$ with random angles. Simulates camera shake. **Hue & Saturation:** Hue offsets $[1 \dots 5]$ and Saturation factors $[0.9 \dots 1.3]$. Simulates color grading or white balance shifts. **Noise (Impulse):** Replacement ratios $[0.002 \dots 0.010]$. Simulates sensor defects or transmission errors. **Resizing (Pixelate & Rescale):** Downsampling ratios $[0.98 \dots 0.90]$ and $[0.90 \dots 0.65]$. Simulates resolution reduction and upscaling artifacts. **Rotation:** Angles $[2° \dots 10°]$. Simulates minor horizon corrections.

*Table 12.* **Comparison with state-of-the-art methods on the proposed CommGen15 dataset.** All methods are evaluated by Acc and AP (%). Following existing methods (Chen et al., 2025; Yan et al., 2025b), we use SDv1.4 from GenImage (Zhu et al., 2023) as the training set.

| Method | Ref | Akool | | ChatGPT | | Doubao | | Flux | | Google Imagen | | Hailuo | | Hunyuan | | Ideogram | |
|---|---|---|---|---|---|---|---|---|---|---|---|---|---|---|---|---|---|
| | | Acc | AP | Acc | AP | Acc | AP | Acc | AP | Acc | AP | Acc | AP | Acc | AP | Acc | AP |
| NPR | CVPR 2024 | 51.3 | 79.3 | 40.3 | 35.9 | 51.6 | 77.3 | 44.0 | 36.7 | 42.3 | 36.2 | 48.4 | 60.5 | 47.5 | 60.7 | 40.9 | 34.6 |
| B-Free | CVPR 2025 | 75.0 | 90.5 | 66.1 | 81.6 | 65.2 | 81.7 | 79.2 | 92.0 | 78.9 | 91.2 | 83.5 | 93.8 | 74.0 | 93.3 | 83.3 | 93.3 |
| AIDE | ICLR 2025 | 66.8 | 97.8 | 67.2 | 97.1 | 72.0 | 98.9 | 73.4 | 98.3 | 71.4 | 98.1 | 65.6 | 97.6 | 68.1 | 98.9 | 67.0 | 98.3 |
| Effort | ICML 2025 | 64.0 | 92.1 | 91.1 | 98.0 | 66.1 | 91.1 | 93.9 | 98.7 | 93.9 | 98.4 | 69.9 | 95.0 | 66.7 | 95.6 | 96.7 | 99.0 |
| DDA | NeurIPS 2025 | 80.9 | 97.0 | 67.9 | 89.2 | 59.3 | 90.7 | 88.2 | 98.3 | 87.1 | 98.0 | 96.5 | 99.6 | 89.0 | 98.9 | 93.4 | 99.1 |
| **Ours** | – | **98.8** | **99.9** | **98.2** | **99.5** | **98.3** | **100.0** | **98.6** | **99.5** | **98.0** | **99.1** | **98.8** | **99.9** | **100.0** | **100.0** | **98.4** | **99.5** |

| Method | Ref | Kling | | Lexica | | Midjourney | | Nano Banana | | Sora | | Stable Diffusion | | Veo | | Mean | |
|---|---|---|---|---|---|---|---|---|---|---|---|---|---|---|---|---|---|
| | | Acc | AP | Acc | AP | Acc | AP | Acc | AP | Acc | AP | Acc | AP | Acc | AP | Acc | AP |
| NPR | CVPR 2024 | 48.4 | 55.0 | 33.3 | 35.9 | 39.6 | 36.1 | 44.9 | 47.9 | 10.9 | 30.9 | 41.0 | 36.2 | 50.3 | 69.2 | 42.3 | 48.8 |
| B-Free | CVPR 2025 | 68.3 | 83.0 | 96.2 | 99.3 | 79.3 | 91.1 | 62.4 | 77.7 | 67.6 | 83.2 | 83.6 | 93.4 | 76.6 | 90.1 | 75.9 | 89.0 |
| AIDE | ICLR 2025 | 65.1 | 97.7 | 76.7 | 98.9 | 67.0 | 97.2 | 66.1 | **97.5** | 72.0 | 98.1 | 66.8 | 97.1 | 65.7 | 96.7 | 68.7 | 97.9 |
| Effort | ICML 2025 | 72.4 | 94.7 | 95.7 | 99.3 | 96.3 | 99.4 | 87.1 | 96.3 | 89.9 | 98.0 | 87.0 | 97.7 | 64.4 | 93.4 | 82.3 | 96.4 |
| DDA | NeurIPS 2025 | 86.1 | 97.9 | 99.0 | 100.0 | 83.6 | 96.4 | 79.0 | 94.0 | 85.2 | 97.3 | 95.7 | 99.5 | 92.8 | 98.8 | 85.6 | 97.0 |
| **Ours** | – | **99.2** | **99.9** | 97.2 | 99.6 | 96.8 | 98.7 | **88.0** | 90.3 | **99.8** | **99.9** | 98.6 | 99.6 | **99.2** | **100.0** | **97.9** | **99.0** |

*Table 13.* **Comparison with state-of-the-art methods on the UniversalFakeDetect dataset.** All methods are evaluated by Acc (%). We follow ForenSynths (Wang et al., 2020) and use the ProGAN-generated images from four categories, namely *car, cat, chair*, and *horse* as the training set while others as the testing sets.

| Method | GAN | | | | | | Deep fakes | Low level | | Perc. loss | | Guided | LDM | | | Glide | | | Dalle | mAcc |
|---|---|---|---|---|---|---|---|---|---|---|---|---|---|---|---|---|---|---|---|---|
| | Pro-GAN | Cycle-GAN | Big-GAN | Style-GAN | Gau-GAN | Star-GAN | | SITD | SAN | CRN | IMLE | | 200 steps | 200 w/cfg | 100 steps | 100 27 | 50 27 | 100 10 | | |
| CNNDet | **100.0** | 85.2 | 70.2 | 85.7 | 79.0 | 91.7 | 53.5 | 66.7 | 48.7 | 86.3 | 86.3 | 60.1 | 54.0 | 55.0 | 54.1 | 60.8 | 63.8 | 65.7 | 55.6 | 69.6 |
| UnivFD | **100.0** | 98.5 | 94.5 | 82.0 | 99.5 | 97.0 | 66.6 | 63.0 | 57.5 | 59.5 | 72.0 | 70.0 | 94.2 | 73.8 | 94.4 | 79.1 | 79.9 | 78.1 | 86.8 | 81.4 |
| LGrad | 99.8 | 85.4 | 82.9 | 94.8 | 72.5 | 99.6 | 58.0 | 62.5 | 50.0 | 50.7 | 50.8 | 77.5 | 94.2 | 95.9 | 94.8 | 87.4 | 90.7 | 89.6 | 88.4 | 80.3 |
| FreqNet | 97.9 | 95.8 | 90.5 | 97.6 | 90.2 | 93.4 | **97.4** | 88.9 | 59.0 | 71.9 | 67.4 | 86.7 | 84.6 | **99.6** | 65.6 | 85.7 | 97.4 | 88.2 | 59.1 | 85.1 |
| NPR | 99.8 | 95.0 | 87.6 | 96.2 | 86.6 | 99.8 | 76.9 | 66.9 | 98.6 | 50.0 | 50.0 | 84.6 | 97.7 | 98.0 | 98.2 | 96.3 | 97.2 | 97.4 | 87.2 | 87.6 |
| DDA | 84.7 | 61.3 | 76.3 | 72.7 | 88.1 | 61.6 | 73.5 | 83.3 | 92.7 | 72.7 | 80.7 | 87.2 | 76.6 | 77.7 | 76.0 | 70.6 | 70.3 | 74.7 | 56.1 | 75.6 |
| AIDE | **100.0** | 98.5 | 84.0 | 99.6 | 73.2 | 99.9 | 54.0 | 69.4 | 70.8 | 60.8 | 60.9 | **88.6** | 98.2 | 97.4 | 98.4 | 98.1 | 98.4 | 97.8 | 97.5 | 86.6 |
| B-Free | 95.5 | 74.5 | 91.5 | 81.3 | 96.6 | 84.8 | 75.2 | **98.9** | 95.9 | 91.7 | 90.5 | 81.9 | 93.8 | 94.0 | 93.9 | 84.3 | 85.8 | 87.3 | 91.7 | 88.9 |
| **Ours** | 97.3 | **99.1** | 96.7 | 77.2 | 98.6 | 90.7 | 77.2 | 54.4 | 91.1 | 88.1 | 88.4 | 87.5 | 92.7 | 98.1 | 97.6 | 97.0 | 96.7 | 97.0 | 97.0 | **90.6** |

*Table 14.* **Impact of varied patch sizes on GenImage (Acc, %).**

| Patch Size | Grid | Token | Midjourney | SDv1.4 | SDv1.5 | ADM | Glide | Wukong | VQDM | BigGAN | Mean |
|---|---|---|---|---|---|---|---|---|---|---|---|
| 14×14 | 16×16 | 256 | 100.0 | 100.0 | 100.0 | 100.0 | 100.0 | 100.0 | 100.0 | 86.7 | **98.3** |
| 28×28 | 8×8 | 64 | 96.6 | 99.6 | 99.7 | 88.3 | 99.2 | 99.5 | 98.0 | 89.8 | 96.3 |
| 56×56 | 4×4 | 16 | 93.6 | 99.4 | 99.4 | 82.9 | 98.6 | 99.0 | 96.5 | 91.1 | 95.0 |
| 112×112 | 2×2 | 4 | 89.7 | 99.5 | 99.5 | 77.0 | 98.1 | 98.9 | 94.5 | 79.4 | 92.1 |

*Table 15.* **Impact of different $\tau$ values on GenImage (Acc, %).**

| $\tau$ | Midjourney | SDv1.4 | SDv1.5 | ADM | Glide | Wukong | VQDM | BigGAN | Mean |
|---|---|---|---|---|---|---|---|---|---|
| 0.25 | 91.5 | 99.3 | 99.4 | 83.3 | 98.6 | 99.2 | 97.5 | 91.6 | 95.1 |
| 0.50 | 100.0 | 100.0 | 100.0 | 100.0 | 100.0 | 100.0 | 100.0 | 86.7 | **98.3** |
| 1.00 | 96.2 | 99.5 | 99.5 | 89.5 | 99.2 | 99.4 | 98.3 | 89.8 | 96.4 |

As shown in Figure 9, PGC exhibits stability. Even under destructive degradations like pixelation or motion blur, the performance drop is minimal compared to the signal loss. This robustness stems from the PGC mechanism: by anchoring on "peak" local patches, the model remains effective as long as a subset of discriminative regions survives the global degradation.

### E.4. Visualization of PGCM

Figures 10 and 11 provide a comprehensive visualization of the Peak-Guided Calibration Module (PGCM) across 15 commercial generators.

**Observation.** A consistent behavioral divergence is observed: **Real Images:** The model's attention (and selected

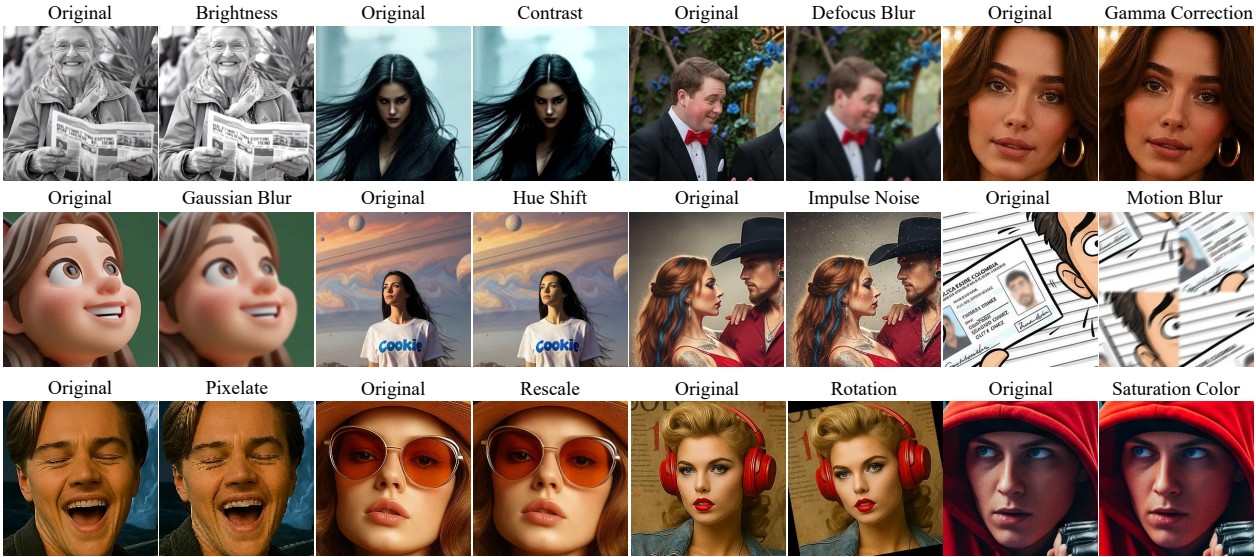

*Figure 8.* **Visualization of Robustness Perturbations.** We apply 12 types of common corruptions to the CommGen15 dataset to evaluate detector stability. Shown here are clean samples alongside their perturbed counterparts.

peak patches) aligns with *semantic foregrounds* (e.g., Horse, objects), which contain the most complex natural high-frequency statistics. **Generated Images:** The attention shifts toward the *background* or peripheral regions.

**Implication.** This visualization validates our core hypothesis: commercial generative models prioritize the fidelity of the main subject to deceive human observers. Consequently, the semantic foreground becomes a "high-fidelity trap" for detectors. By calibrating the decision based on peak patches (often located in the imperfect background), PGC effectively bypasses this trap, detecting artifacts that would otherwise be masked by the realistic subject.

## F. Limitations and Future Work

The core contribution of this work is the PGC framework, which effectively addresses the critical challenge where subtle forgery traces are masked by high-fidelity content. Through extensive experiments on our proposed Comm-Gen15 benchmark and standard datasets, PGC has demonstrated superior generalization and robustness against commercial model forgeries compared to existing state-of-the-art methods. However, a potential limitation of our current framework lies in its binary training objective, which unifies all diverse forgery types into a single "Fake" category. While this approach simplifies the decision boundary, it may inadvertently overlook the fine-grained artifact fingerprints specific to different generative architectures, potentially underutilizing the unique spectral or statistical discrepancies inherent to specific synthesis methods.

In the future, we plan to extend the "peak-guided" philoso-

phy from the spatial domain to the temporal domain. Given that video generation models (e.g., Sora, Veo) are rapidly emerging, we aim to design a temporal PGC module that identifies "peak flickering" or localized motion inconsistencies across frames, thereby addressing the growing threat of AI-generated videos. Furthermore, we hope our strategy of explicitly isolating local anomalies can inspire applications in broader fields, such as medical image anomaly detection and face anti-spoofing.

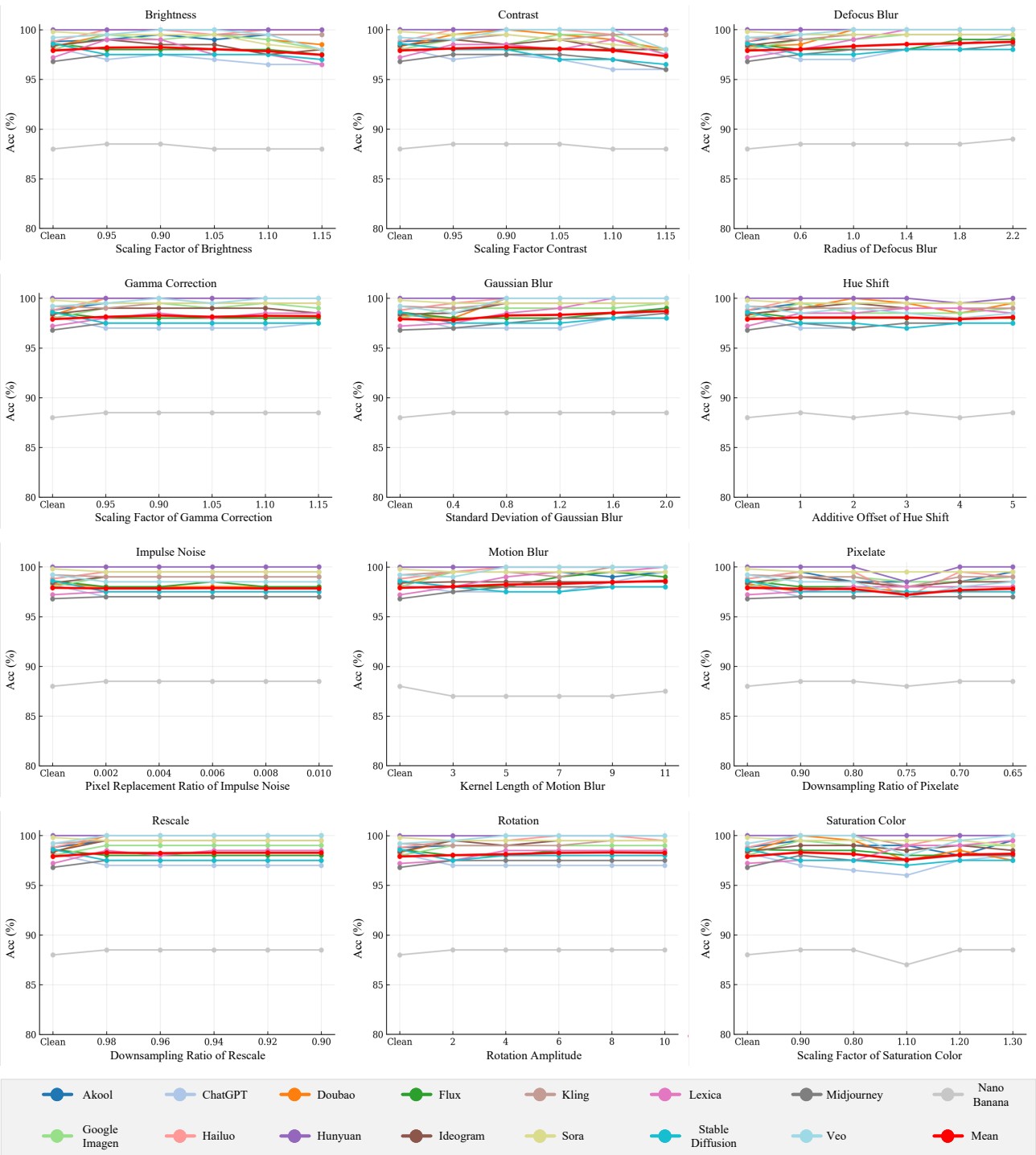

*Figure 9.* **Robustness on CommGen15 under common image perturbations.** We train our detector on SDv1.4 and report its Acc on CommGen15 under 12 families of test-time perturbations with increasing severity. "Clean" denotes evaluation on the unperturbed inputs. For each perturbation, we sweep five severity levels following the parameter settings described in Sec. E.3 (e.g., scaling factors for brightness/contrast/saturation and gamma, blur radius or standard deviation for defocus/gaussian blur, kernel length for motion blur, hue offsets in HSV space, replacement ratio for impulse noise, downsampling ratios for pixelation/rescaling, and rotation magnitudes with center cropping). Overall, the Acc curves remain largely stable as the perturbation strength increases, indicating strong robustness to typical capture degradations and mild post-processing.

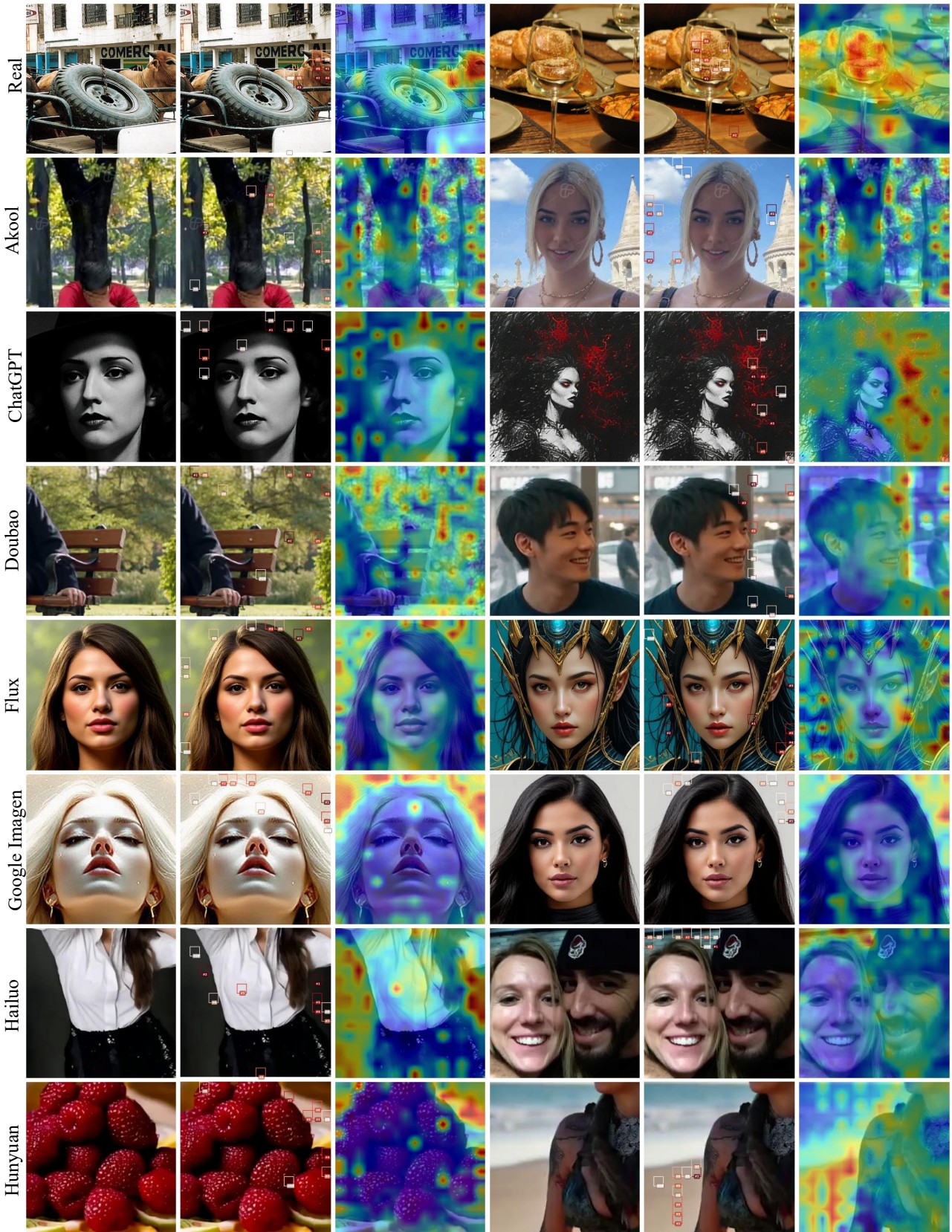

*Figure 10.* **Visualization of PGC (Part 1/2).** We visualize the selected "peak patches" (red boxes) and decision heatmaps. Note the consistent shift: Real images trigger responses on the semantic foreground, while generated images (across diverse platforms like Akool, ChatGPT, Flux) trigger responses in the background/peripheral regions.

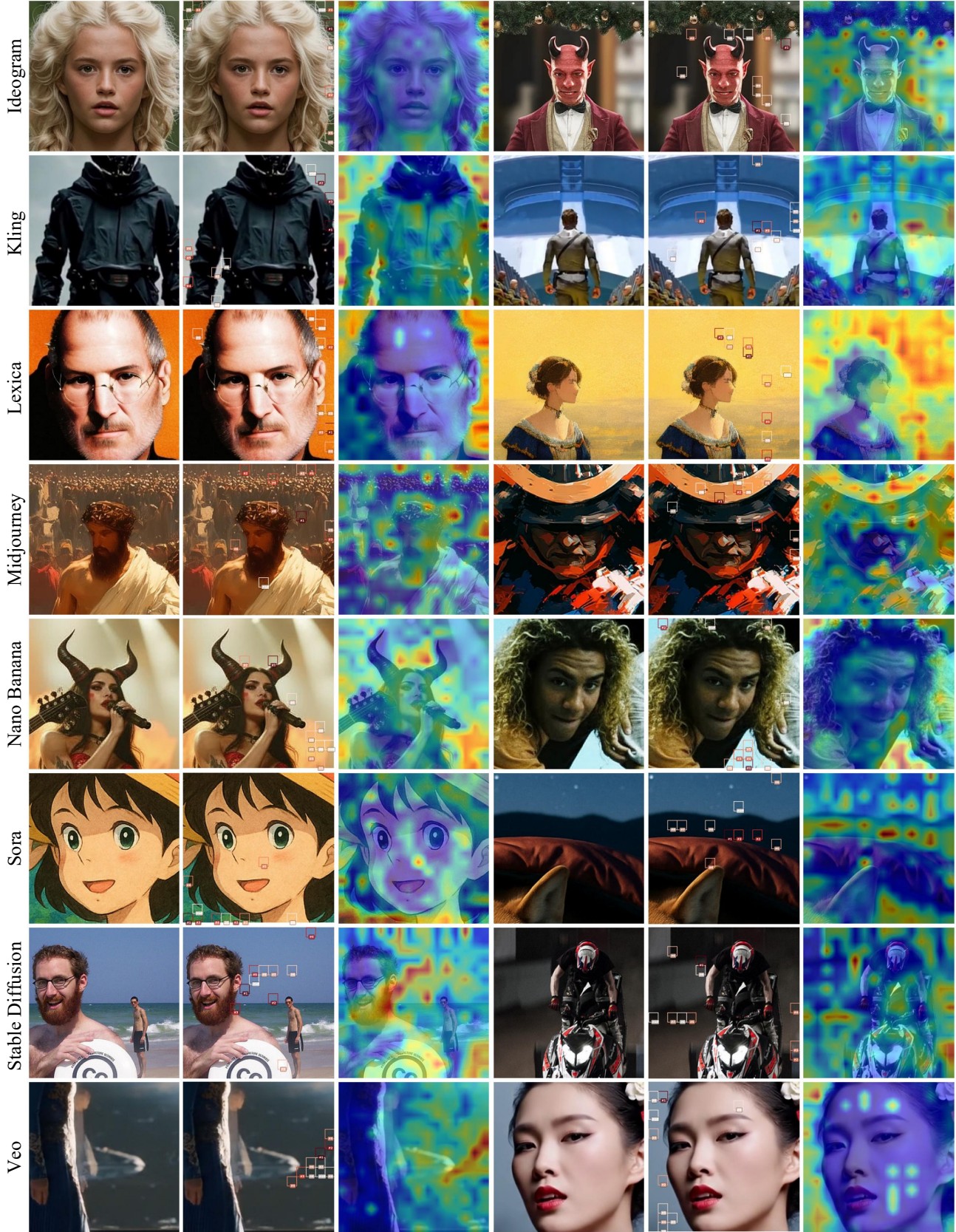

*Figure 11.* **Visualization of PGC (Part 2/2).** Continued visualization across platforms like Midjourney, Sora, and Veo. The mechanism explicitly targets background artifacts that survive high-fidelity synthesis.

