# OpenReview forum: "PGC: Peak-Guided Calibration for Generalizable AI-Generated Image Detection"
_ICML.cc/2026/Conference — ICML 2026 regular_

### Official Review · Reviewer_2QbL · 2026-02-23

**Soundness:** 3
**Presentation:** 3
**Significance:** 3
**Originality:** 3
**Overall Recommendation:** 4
**Confidence:** 4

**Summary:**

This paper studies generalization in AI-generated image detection. The authors argue that modern generators focus heavily on foreground regions, which can hide subtle artifacts in the background. As a result, detectors based on global features may miss these fine-grained traces. To address this, the paper proposes Peak-Guided Calibration (PGC). The method extracts high-response local patches and uses them to adjust the global representation. It combines a dual-stream design with soft peak aggregation and logit-level calibration. Experiments on multiple datasets, including commercial generative models, show improved generalization over recent baselines.

**Compliance With Llm Reviewing Policy:**

Affirmed.

**Final Justification:**

The rebuttal adequately addressed my main concerns. In particular, the additional comparisons with simpler alternatives, the SHAP-based foreground/background analysis, the clarification regarding robustness beyond saturated benchmarks, and the added results on traditional manipulation settings all strengthen the paper and make the technical motivation more convincing. Overall, I maintain a positive assessment of the submission.

**Key Questions For Authors:**

Please refer to Weakness.

**Limitations:**

It appears to me that the evaluation primarily focuses on modern generative models. It would be helpful to understand how the method performs on more traditional manipulation settings, such as face swap or attribute editing, where artifacts are concentrated in foreground regions rather than background areas.

**Strengths And Weaknesses:**

Strengths
1. The paper addresses the generalization challenge in AI-generated image detection, which is both important and practically relevant. The observation that modern generative models allocate capacity unevenly across spatial regions is intuitive and aligns with recent trends in high-fidelity foreground rendering. Framing detection failure as a consequence of semantic dominance provides a coherent narrative for the proposed method.

2.  The Peak-Guided Calibration framework is cleanly organized (although I feel Eq. 5 needs more details about each tem). The dual-stream design, peak-based local extraction, and global calibration are logically connected. The implementation appears straightforward and reproducible. The method does not rely on heavy architectural changes or unstable optimization tricks, which increases its practical appeal.

3.	The paper evaluates across multiple datasets and includes tests on commercial generative models. The improvements are consistent across settings and show meaningful gains in cross-model generalization. The inclusion of ablations and visualizations helps support the claims and demonstrates that the authors conducted substantial experimental work.



Weaknesses

1.	While the method is clean and effective, the core mechanism (Eq. 3), soft peak-based aggregation combined with logit-level calibration, resembles feature reweighting or selective pooling strategies seen in prior work. The paper would benefit from a stronger argument explaining whether this constitutes a new representational principle or primarily a careful engineering refinement at a specific layer.

2. The motivation relies on the notion of “semantic dominance” and foreground bias. However, the paper does not provide a deeper quantitative analysis of attention allocation, representation geometry, or information flow to rigorously support this claim. A more formal examination of why peak-guided calibration outperforms simpler alternatives (e.g., max pooling, standard attention reweighting, or spatial masking) would strengthen the contribution.

3.	The empirical gains over strong baselines are relatively modest in some settings. For example, in Table 2 the improvement over CoD (96.2/99.4) is incremental, and in Table 3 the margins over Effort and DDA are also small. Given that these baselines already achieve performance close to saturation (around 98% or higher), it would be helpful to better understand whether the reported improvements reflect genuine robustness gains or sensitivity to dataset-specific characteristics. Additional evaluations would increase confidence that the method is not inadvertently overfitting to particular artifact distributions.

---

> ### Author Rebuttal · Authors · 2026-03-31
>
> We sincerely thank the reviewer for recognizing: **"...The implementation appears straightforward and reproducible. The method does not rely on heavy architectural changes or unstable optimization tricks, which increases its practical appeal ... the authors conducted substantial experimental work"** We will revise Eq. (5) to explicitly define each term. Our responses are as follows.
>
> **Q1. Analysis on patch processing strategies (principle vs. refinement):**
>
> We clarify that PGC is not a generic engineering refinement, but a task-specific mechanism for AIGI detection, motivated by a fundamental generation bias: modern generators prioritize caption-described foregrounds, leaving backgrounds weakly constrained and thus more artifact-prone.
>
> Compared with prior patch selection methods: (1) PGC operates on deep feature maps rather than cropped patches, preserving global context. (2) Unlike Hard Top-K that abruptly discards features, our soft-peak aggregation amplifies localized artifacts while preserving the continuous contextual landscape (mAcc drops from 98.3% to 96.4% if replaced by Top-K, proving its structural necessity).
>
> | Strategy | Midjourney | SD_v1.4 | SD_v1.5 |  ADM  | GLide | Wukong | VQDM  | BigGAN | mAcc |
> | :------- | :--------: | :-----: | :-----: | :---: | :---: | :----: | :---: | :----: | :--: |
> | Top-K    |    96.3    |  96.5   |  96.4   | 96.2  | 96.2  |  96.6  | 96.3  |  96.6  | 96.4 |
> | Ours     |   100.0    |  100.0  |  100.0  | 100.0 | 99.9  | 100.0  | 100.0 |  86.7  | 98.3 |
>
>
> **Q2. Motivation and alternatives:**
>
> To validate the foreground bias and analyze the decision information flow, we performed a **SHAP-based attribution analysis**. Using SAM-generated masks, we measured the density of positive ($d^+$) and negative ($d^-$) SHAP values to account for the area mismatch between the foreground (FG) and background (BG). When preserving both positive and negative evidence, the conclusion holds: the evaluated models rely on background cues rather than foreground content. Specifically, the **background provides stronger positive evidence than the foreground**, with $d^{+}(\mathrm{BG}) - d^{+}(\mathrm{FG}) > 0$ across all models (mean gap: **0.307**). Moreover, within the foreground, positive evidence is even slightly weaker than negative evidence ($d^{+}(\mathrm{FG}) - d^{-}(\mathrm{FG}) < 0$, mean: **-0.033**). These quantitative results confirm that the foreground does not serve as a primary positive contributor and may even provide weak counter-evidence.
>
> Guided by this background-dominant information flow, we can explain why peak-guided calibration relies on scattered cues and thus outperforms simpler alternatives. Compared with standard reweighting, which gives only marginal improvement, peak-focused calibration is more effective: hard Top-K reaches **96.4%** mACC, and our soft peak aggregation further improves it to **98.3%**.
>
> | Method / Strategy                          | mACC     |
> | ------------------------------------------ | -------- |
> | Global Baseline _(No Calibration)_         | 92.1     |
> | Spatial Masking _(Random Patches)_         | 93.7     |
> | Standard Reweighting _(Dense Gating)_      | 89.8     |
> | Standard Reweighting _(Concat + MLP)_      | 93.5     |
> | Max Pooling _(Hard Top-K Selection)_       | 96.4     |
> | **Ours: Soft Peak Aggregation (Additive)** | **98.3** |
>
> **Q3. Generalization vs. dataset sensitivity**
>
> The modest gains on GenImage are due to **performance saturation (at ~98%)**, rather than dataset sensitivity. Our genuine generalization is evidenced by three facts:
>
> - (1) Scale of Evaluation: Evaluated across **>50 diverse generative models**, this extensive cross-domain testing rules out the possibility of overfitting to specific artifact distributions.
>
> - (2) +12.3% Gain In-the-Wild: When moving from saturated datasets to the challenging in-the-wild **CommGen15** benchmark, PGC reveals its true superiority, outperforming the strongest baseline (DDA) by **+12.3%**.
>
> - (3) Robustness: As shown in Fig. 6, PGC consistently maintains **>97% ACC** across various corruptions. In stark contrast, strong baselines (DDA, Effort) degrade significantly, dropping to **below 75%** under shot noise.
>
> **Q4. Performance on traditional manipulation:**
>
> This setting is already included in **Table 4**. PGC achieves the best **mACC (82.1%)**, outperforming DDA (**80.5%**) on traditional manipulation benchmarks, showing that it also handles foreground-localized artifacts.
>
> | Method | BlendFace | E4S  | FaceSwap | InSwap | SimSwap | mACC |
> | ------ | --------- | ---- | -------- | ------ | ------- | ---- |
> | SAFE   | 47.3      | 47.6 | 50.7     | 49.7   | 49.0    | 48.9 |
> | DDA    | 77.9      | 82.3 | 77.3     | 82.0   | 83.3    | 80.5 |
> | Ours   | 71.8      | 88.5 | 87.0     | 80.4   | 82.7    | 82.1 |

---

> > ### Author Rebuttal · Reviewer_2QbL · 2026-04-02
> >
> > Thanks for the detailed rebuttal. My main concerns have been adequately addressed. In particular, the additional comparisons with simpler alternatives, the SHAP-based quantitative analysis of foreground/background evidence, the clarification regarding robustness beyond saturated benchmarks, and the results on traditional manipulation settings all strengthen the paper. I also very much appreciate the additional analysis provided in the rebuttal.

---

> > > ### Author Response · Authors · 2026-04-02
> > >
> > > We appreciate your time reviewing our rebuttal and your positive response. We are glad that the additional comparisons, SHAP analysis, and robustness results have addressed your concerns.

---

### Official Review · Reviewer_ube3 · 2026-03-08

**Soundness:** 2
**Presentation:** 2
**Significance:** 2
**Originality:** 2
**Overall Recommendation:** 4
**Confidence:** 4

**Summary:**

The authors argue that generative models primarily prioritize subject semantics, which makes discriminative forgery traces more likely to appear in limited background or boundary regions. However, conventional global representations are often dominated by semantic content, thereby obscuring these fine-grained forensic cues. To address this issue, the paper proposes Peak-Guided Calibration, which first assigns scores to local patches, then performs peak aggregation to extract local evidence, and finally combines this evidence with global features to form the final representation.

**Compliance With Llm Reviewing Policy:**

Affirmed.

**Final Justification:**

The authors provided a thorough rebuttal addressing most technical concerns

**Key Questions For Authors:**

a) As shown in Fig.1, the baseline already tends to attend to background regions in AI-generated images, and similar background-focused patterns are also observed after training (see the appendix). Therefore, it would be valuable to clarify whether PGC primarily functions as a finer-grained local evidence reweighting (or peak-based pooling) mechanism that emphasizes a small number of high-response patches.
b) This method can essentially be regarded as a patch selection strategy. Given that a substantial body of prior work has explored patch selection mechanisms (e.g., AIDE[1] and PatchCraft[2]), a systematic ablation study across different selection strategies would more clearly demonstrate the effectiveness and relative advantages of the proposed design.
c) Table 3 indicates that the method proposed in this paper is trained on SDv1.4. Do the other methods follow the same training protocol as well?

**Limitations:**

Yes

**Strengths And Weaknesses:**

Strengths
a) The core problem is identified as semantic dominance, which overwhelms forgery clues, and a peak patch calibration method is proposed to address this issue.
b) Extensive experiments demonstrate that PGC achieves state-of-the-art performance.

Weaknesses
a) As shown in Fig.1, the baseline already tends to attend to background regions in AI-generated images, and similar background-focused patterns are also observed after training (see the appendix). Therefore, it would be valuable to clarify whether PGC primarily functions as a finer-grained local evidence reweighting (or peak-based pooling) mechanism that emphasizes a small number of high-response patches.
b) This method can essentially be regarded as a patch selection strategy. Given that a substantial body of prior work has explored patch selection mechanisms (e.g., AIDE[1] and PatchCraft[2]), a systematic ablation study across different selection strategies would more clearly demonstrate the effectiveness and relative advantages of the proposed design.
c) Table 3 indicates that the method proposed in this paper is trained on SDv1.4. Do the other methods follow the same training protocol as well?


Reference
[1] Yan S, et al. A sanity check for ai-generated image detection. ICLR 2025
[2] Zhong N, et al. Patchcraft: Exploring texture patch for efficient ai-generated image detection. arXiv 2023.

---

> ### Author Rebuttal · Authors · 2026-03-31
>
> We sincerely thank the reviewer for acknowledging that our PGC framework achieves state-of-the-art performance. We address your insightful questions below.
>
> **Q1. The clarification of Fig. 1 and the role of PGC:**
>
> We thank the reviewer for pointing out the confusing caption in Fig. 1. We clarify that the heatmaps visualize the results of our PGC, not the baseline. **Specifically, PGC extracts and upweights peak patches, utilizing the resulting peak artifact features to calibrate the global representation.** This targeted calibration is necessary because global representations are easily dominated by foregrounds. To verify this, we conducted a SHAP-based attribution analysis (using SAM masks for area normalization). The results confirm that across several generators, the background provides stronger positive evidence for fake detection than the foreground (mean: +0.307). In contrast, the high-fidelity foreground acts as a distractor, yielding net negative evidence (mean: -0.033). These attribution findings expose a critical gap in the field and clarify the exact role of PGC:
>
> * **(1) The Baseline Flaw:** Existing detectors rely on global representations that are inherently dominated by these high-energy, distracting foregrounds. As a result, critical background artifacts are overshadowed. Table 3 confirms this masking effect: SOTA global methods (e.g., Effort, DDA) suffer significant performance drops on high-fidelity CommGen15, dropping to a mean fake accuracy (F.Acc) of only **66.7%** and **72.6%**.
>
> * **(2) The Essential Role of PGC:** Rather than simply tweaking global attention, PGC is designed to proactively aggregate salient local clues (peak patches) to explicitly break the foreground domination. By "rescuing" these submerged background artifacts to calibrate the global decision, PGC significantly boosts the mean F.Acc to **95.7%** (Table 3).
>
> We will update the caption of Fig. 1 to  specify the visualization source, and clarify this motivation in the paper.
>
>
> **Q2. Analysis on patch selection strategies:**
>
> While sharing the intuition of utilizing local patches, PGC differs from AIDE and PatchCraft in three aspects:
>
> - **(1) Strategy Differences:** AIDE and PatchCraft perform non-differentiable hard cropping on raw images, disrupting global semantics. In contrast, PGC operates on deep spatial feature maps using continuous soft aggregation, fully preserving the global receptive field.
>
> * **(2) Overall Superiority:** Benefiting precisely from this feature-level design, when evaluated using the exact same training data, PGC significantly outperforms both PatchCraft (89.4% to 98.3%) and AIDE (87.1% to 98.3%) in mAcc.
>
> * **(3) Ablation of patch selection (Soft vs. Hard Selection):** To isolate and prove the effectiveness of our specific selection mechanism, we replaced our soft aggregation with a Hard Top-K selection. As shown below, this degradation from soft to hard selection drops the mAcc from 98.3% to 96.4%, proving that our method balances local peak amplification and global gradient preservation.
>
> | Strategy   | Midjourney | SD_v1.4 | SD_v1.5 | ADM   | GLide | Wukong | VQDM  | BigGAN | mACC     |
> | ---------- | ---------- | ------- | ------- | ----- | ----- | ------ | ----- | ------ | -------- |
> | PatchCraft | 89.7       | 95.0    | 94.6    | 81.6  | 83.5  | 90.9   | 88.2  | 91.5   | 89.4     |
> | AIDE       | 81.4       | 99.8    | 99.8    | 78.5  | 91.8  | 98.9   | 80.2  | 66.8   | 87.1     |
> | Ours       | 100.0      | 100.0   | 100.0   | 100.0 | 99.9  | 100.0  | 100.0 | 86.7   | **98.3** |
> | Top-K      | 96.3       | 96.5    | 96.4    | 96.2  | 96.2  | 96.6   | 96.3  | 96.6   | 96.4     |
>
>
> **Q3. Training protocol alignment in Table 3:**
>
> Yes. **As stated in Section 5.1**, the methods in Tables 2 and 3 follow the same SDv1.4 training protocol as ours, except B-Free and DDA, whose original designs require paired reference images.

---

> > ### Author Rebuttal · Reviewer_ube3 · 2026-04-03
> >
> > The authors provided a thorough rebuttal addressing most technical concerns

---

> > > ### Author Response · Authors · 2026-04-03
> > >
> > > We sincerely appreciate your recognition of our rebuttal and your insightful comments, which have greatly helped improve the quality of our manuscript. We are also very grateful for your kind upward adjustment of the review score.

---

### Official Review · Reviewer_vG72 · 2026-03-12

**Soundness:** 3
**Presentation:** 3
**Significance:** 2
**Originality:** 2
**Overall Recommendation:** 5
**Confidence:** 4

**Summary:**

This paper introduced an AI-generated image detector framework called Peak-Guided Calibration (PGC). PGC operates through aggregating salient features and employing a peak-sensitive aggregation for local clues, which calibrates towards a global decision. PGC can then use the location of discriminative traces within the image to identify AI-generated content. Empirical results demonstrates that PGC reaches superior performance when compared with other state-of-the-art models across different datasets.

**Compliance With Llm Reviewing Policy:**

Affirmed.

**Final Justification:**

The authors have addressed my concerns, I have increased my score.

**Key Questions For Authors:**

1. Is there a theoretical analysis on the key assumption of this proposed method, which is AI-generated artifacts will appear as localized peak patches? I saw the empirical demonstration and results that this might be true for current datasets and models, but limited theoretical discussion/justification on why the authors decided to focus on local patches.

2. Did the author test robustness against better perturbations such as adversarial perturbation or adaptive attacks?

**Limitations:**

yes

**Strengths And Weaknesses:**

Strengths:

1. This paper proposed a solution to an important problem in generated image forensic, which could be useful to modern applications such as synthetic media detection. The proposed method is well-motivated, with the authors pointing out that existing detectors rely on global representations and PGC addressed this limitation.

2. On top of proposing the method, this paper also introduced CommGen15, which is a benchmark dataset composed of both images and videos sampled from 15 commercial models. The purpose of introducing this was to ensure their methods is tested on the current state-of-the-art model generated images, evaluating against true threats that are up to date.

3. This paper demonstrated superior performance of the proposed PGC method, including both detection accuracy and generalization across different commercial models. Generalization is important in this case given the diverse selection of models that exist in the real-world case. And as demonstrated in Figure 3, the proposed PGC method is able to support such diversity better than current existing methods.

4. The writing of this paper is clear and the logic is easy to follow. The overview figure is neatly organized and the figures in experimental section demonstrates key performances well.

Weaknesses:

1. The robustness analysis seems limited, it seems like only brightness changes, blurring, and noise to the image were tested.

---

> ### Author Rebuttal · Authors · 2026-03-31
>
> We sincerely thank the reviewer for recognizing: **"This paper proposed a solution to an important problem ... is well-motivated ... demonstrated superior performance ... generalization across different commercial models ... The writing of this paper is clear and the logic is easy to follow ... The overview figure is neatly organized and the figures in experimental section demonstrates key performances well."** We have thoroughly addressed your insightful suggestions below.
>
>
> **Q1. Theoretical justification of localized peak artifacts:**
>
> In modern image generative models, generation quality is often spatially non-uniform. Training captions usually describe foreground objects much more explicitly than backgrounds, and cross-attention therefore focuses much more strongly on foreground regions [1,2]. As a result, prompt-relevant foreground content is optimized to be closer to natural images, while background and transition regions are only weakly constrained. This leaves larger residual errors in those regions, making generation traces more likely to stand out in a few local background patches. A simple way to formalize this intuition is through the following optimization problem:
>  $$
> \min_{\{c_i\}} L_{\mathrm{gen}}=\sum_i w_i\, d_i(c_i)
> \qquad \text{s.t.} \qquad \sum_i c_i \le C.
>  $$
> Here, $i$ indexes image patches, $w_i$ denotes the conditioning strength of patch $i$, $c_i$ is the effective generation capacity allocated to that patch, $d_i(c_i)$ denotes the residual generation error of patch $i$, and $C$ is the total capacity budget of the generator. Since $d_i(c_i)$ decreases as more capacity is assigned, minimizing this objective naturally allocates more capacity to strongly conditioned regions with larger $w_i$. Therefore, foreground regions tend to be better optimized, while weakly conditioned background and transition regions receive less capacity and retain larger residual errors, making generation traces more likely to appear as localized high-response patches there.
>
> References
>
> [1] Tao Yang, Cuiling Lan, Yan Lu, and Nanning Zheng. Diffusion Model with Cross Attention as an Inductive Bias for Disentanglement. NeurIPS, 2024.
>
> [2] Ryugo Morita, Stanislav Frolov, Brian Bernhard Moser, Takahiro Shirakawa, Ko Watanabe, Andreas Dengel, and Jinjia Zhou. TKG-DM: Training-free Chroma Key Content Generation Diffusion Model. CVPR, 2025: 13031-13040.
>
>
> **Q2. Robustness to adversarial perturbations:**
>
> In real-world deployment scenarios, the architecture and parameters of our detection algorithm are kept confidential, making black-box attacks relevant threat. To simulate this, we evaluated PGC against black-box FGSM perturbations on 1,600 samples across 8 generation platforms. As shown below, under mild perturbations ($\epsilon \le 0.08$), the mean accuracy (mACC) remains highly stable (maintaining $>98\%$). Even under a relatively strong attack ($\epsilon=0.16$), the mACC is well preserved at 96.5%. These results demonstrate that PGC exhibits robustness against realistic adversarial evasion attempts.
>
> | $\epsilon$ | chatgpt-image | flux | google-imagen | hailuo | hunyuan | kling | nano_banana | sora  | mACC |
> | ---------- | ------------- | ---- | ------------- | ------ | ------- | ----- | ----------- | ----- | ---- |
> | 0          | 99.5          | 98.5 | 97.5          | 99.5   | 100.0   | 100.0 | 89.5        | 100.0 | 98.1 |
> | 0.04       | 99.5          | 99.0 | 98.5          | 99.5   | 100.0   | 100.0 | 89.5        | 100.0 | 98.3 |
> | 0.08       | 99.5          | 99.0 | 98.0          | 99.5   | 100.0   | 100.0 | 90.0        | 99.5  | 98.2 |
> | 0.12       | 99.0          | 98.5 | 97.5          | 99.0   | 100.0   | 100.0 | 89.0        | 99.5  | 97.8 |
> | 0.16       | 97.5          | 96.5 | 96.0          | 98.5   | 100.0   | 98.5  | 87.5        | 97.5  | 96.5 |

---

> > ### Author Rebuttal · Reviewer_vG72 · 2026-04-01
> >
> > Thank you for the rebuttal. My concern remains for a stronger robustness evaluation. While the results under FGSM seems promising, FGSM is a now considered a relatively weak, single-step attack. Stronger black-box attacks, such as transfer-based or query-based methods that approximate iterative attacks like Projected Gradient Descent (PGD) [1], are commonly used as a more rigorous robustness benchmark. It would strengthen the paper to include such results.
> > References:
> > 1. Madry, Aleksander, et al. "Towards deep learning models resistant to adversarial attacks." arXiv preprint arXiv:1706.06083 (2017).

---

> > > ### Author Response · Authors · 2026-04-02
> > >
> > > We thank the reviewer for pointing out the necessity of a stronger robustness benchmark. **We conducted adversary attacks to our detector with PGD as you suggested, and the results demonstrate that our method effectively resists transfer-based black-box attacks.** We adopt the black-box attacks because, in realistic scenarios, the detector is unlikely to be exposed to attackers. To validate this, we evaluated our approach against 40-step PGD attacks generated by two surrogate models: ResNet-50, a standard baseline, and Effort [1] (Yan et al., ICML 2025, Oral), which is architecturally more similar to our model and therefore serves as a stronger surrogate. Both models were trained on the same SDv1.4 dataset as our model, with the perturbation budget $\epsilon$ ranging from 0.2 to 0.5.
> > >
> > > As shown in the table, our method maintains over 98% mAcc under $\epsilon \le 0.3$. Interestingly, we observed that adding a small perturbation (to both real and AI-generated images, $\epsilon=0.2$) slightly improves the detection Acc compared to clean images ($\epsilon=0$), and a similar phenomenon is also observed in FGSM. The reason could be that the adversarial noise introduces randomness to amplify the artifacts of AI-generated images, making these synthetic patterns more distinguishable to our detector. Under the constraint of $\epsilon=0.5$, the mAcc drops to ~90% against the Effort model, but remains at 95.9% against ResNet-50 due to differences in feature representations. Importantly, as noted in the original PGD paper [2], an extreme perturbation of $\epsilon \ge 0.5$ allows an adversary to construct a uniformly gray image and fool any classifier. Therefore, sustaining ~90% Acc under a perturbation budget of $\epsilon=0.5$ demonstrates the robustness of our method.
> > >
> > > | Surrogate Model | Epsilon $\epsilon$ | ChatGPT | Flux  | Google Imagen | Hailuo | Hunyuan | Kling | Nano Banana | Sora  | mAcc |
> > > | ------------------ | ------- | ------- | ----- | ------------- | ------ | ------- | ----- | ----------- | ----- | ---- |
> > > | -                  | 0       | 99.5    | 98.5  | 97.5          | 99.5   | 100.0   | 100.0 | 89.5        | 100.0 | 98.1 |
> > > | Effort             | 0.2     | 100.0   | 99.0  | 99.0          | 100.0  | 99.5    | 99.5  | 98.5        | 99.5  | 99.4 |
> > > | Effort             | 0.3     | 98.5    | 98.5  | 98.5          | 98.5   | 99.0    | 99.0  | 98.0        | 97.5  | 98.4 |
> > > | Effort             | 0.4     | 95.5    | 95.5  | 94.0          | 96.0   | 96.0    | 95.5  | 97.0        | 94.0  | 95.4 |
> > > | Effort             | 0.5     | 90.0    | 92.0  | 90.5          | 91.5   | 90.5    | 89.0  | 89.0        | 88.5  | 90.1 |
> > > | Resnet50           | 0.2     | 100.0   | 100.0 | 100.0         | 100.0  | 100.0   | 100.0 | 100.0       | 99.0  | 99.9 |
> > > | Resnet50           | 0.3     | 99.5    | 100.0 | 100.0         | 99.5   | 99.0    | 100.0 | 100.0       | 99.0  | 99.6 |
> > > | Resnet50           | 0.4     | 99.0    | 99.5  | 98.5          | 99.0   | 97.5    | 99.0  | 100.0       | 98.0  | 98.8 |
> > > | Resnet50           | 0.5     | 97.0    | 97.5  | 96.0          | 96.5   | 94.5    | 94.5  | 97.5        | 93.5  | 95.9 |
> > >
> > > **References**
> > >
> > > [1] Zhiyuan Yan, et al. "Orthogonal Subspace Decomposition for Generalizable AI-Generated Image Detection," ICML, 2025.
> > >
> > > [2] Aleksander Madry, et al. "Towards Deep Learning Models Resistant to Adversarial Attacks," ICLR, 2018.

---

### Official Review · Reviewer_72DQ · 2026-03-13

**Soundness:** 3
**Presentation:** 3
**Significance:** 3
**Originality:** 3
**Overall Recommendation:** 3
**Confidence:** 5

**Summary:**

This paper introduces Peak-Guided Calibration (PGC), a novel framework designed to enhance the generalization capabilities of AI-generated image detection systems when applied to high-fidelity or previously unseen generation models. Motivated by the observation that global feature representations often fail to capture subtle local artifacts, PGC incorporates a peak-guided aggregation mechanism that identifies highly discriminative local patches and leverages them to refine global predictions. Furthermore, this work presents CommGen15, a newly developed benchmark dataset comprising images generated by various commercial models, aimed at emulating real-world conditions more effectively. Experimental evaluations conducted on multiple public datasets as well as CommGen15 demonstrate consistent performance gains compared to existing approaches, underscoring the efficacy of the proposed calibration strategy.

**Compliance With Llm Reviewing Policy:**

Affirmed.

**Key Questions For Authors:**

1. The motivation is largely based on the CAM visualization in Fig.1, where the authors conclude that real images trigger attention on the foreground while generated images shift attention to the background. However, CAM is normalized and mainly highlights positive activations, while negative contributions are not shown. Could the authors clarify whether the conclusion still holds when using attribution methods that consider both positive and negative evidence?
2. The paper claims that the high-fidelity foreground acts as a semantic distractor that obscures discriminative traces. An alternative explanation is that complex foreground objects may actually introduce more generation artifacts, making them easier rather than harder to detect. Can the authors provide quantitative evidence showing that discriminative signals are indeed weaker in foreground regions than in background regions?
3. The calibration module combines local peak scores with the global logit using a simple additive formulation. It is unclear whether the improvement comes from the peak selection itself or simply from logit re-weighting. Have the authors compared alternative fusion strategies to justify the specific design choice?

**Limitations:**

yes

**Strengths And Weaknesses:**

**Strengths:**
This study addresses a critical challenge in computer vision: enhancing the generalization capabilities of AI-generated image detection under high-fidelity conditions and unseen generative models. The proposed Peak-Guided Calibration framework demonstrates simplicity, intuitiveness, and consistent performance improvements across diverse benchmarks. Furthermore, the introduction of the CommGen15 dataset provides significant value by enabling evaluations tailored to real-world scenarios involving commercial generative models. Experimental results are comprehensive, encompassing comparisons with state-of-the-art methods, ablation studies, robustness analyses, and visualizations that collectively substantiate the empirical findings.

**Weaknesses:**
The primary motivation hinges on CAM-based visualizations; however, these only capture normalized positive activations without conclusively supporting the hypothesis that foreground semantics suppress discriminative traces. Additionally, the assumption that artifacts predominantly concentrate within peak patches remains insufficiently validated across various generator architectures. The calibration mechanism employs an additive fusion design which appears heuristic in nature with limited theoretical justification provided for its formulation.

---

> ### Author Rebuttal · Authors · 2026-03-31
>
> We sincerely thank the reviewer for recognizing: "**This study addresses a critical challenge .... CommGen15 dataset provides significant value ... framework demonstrates simplicity, intuitiveness, and consistent performance improvements across diverse benchmarks ... Experimental results are comprehensive...**". We appreciate your feedback and are committed to solving all concerns.
>
> **Q1. Considering both positive and negative evidence:**
>
> When using an attribution method that preserves both positive and negative evidence, our conclusion still holds: the evaluated models rely more on background cues than on foreground content.
>
>  We performed a **SHAP-based attribution analysis**. Using SAM-generated foreground/background (FG/BG) masks, we measured attribution density to account for the area mismatch between the two regions. Specifically, for each region, we quantified the densities of positive and negative SHAP values, where positive values contribute positively to the model decision and negative values contribute negatively. The same trend holds consistently across all evaluated generators. In particular, the **background provides stronger positive evidence than the foreground** for the evaluated models, with $d^{+}(\mathrm{BG}) - d^{+}(\mathrm{FG}) > 0$ in every case (mean: **0.307**). Moreover, within the foreground, positive evidence is slightly weaker than negative evidence, as $d^{+}(\mathrm{FG}) - d^{-}(\mathrm{FG}) < 0$ for all models (mean: **-0.033**). These results indicate that the models rely more on background cues, whereas the foreground does not serve as a primary positive contributor and may even provide weak counter-evidence.
>
> | Metric                          | Doubao | Akool  | Flux   | Google Imagen | Hailuo | Nano Banana | Stable Diffusion | Mean   |
> | ------------------------------- | ------ | ------ | ------ | ------------- | ------ | ----------- | ---------------- | ------ |
> | BG–FG positive <br>evidence gap | 0.244  | 0.244  | 0.275  | 0.409         | 0.273  | 0.29        | 0.411            | 0.307  |
> | FG net evidence                 | -0.045 | -0.027 | -0.033 | -0.034        | -0.015 | -0.039      | -0.039           | -0.033 |
>
>
>
> **Q2. Verification of the key claim:**
>
> Through a quantitative perturbation-based masking experiment, we reveal that discriminative signals are indeed weaker in foreground regions than in background regions.
>
> This experiment compares the effects of foreground and background on the detector's decision. In the table below, FG and BG denote the increase in fake probability after blurring the foreground or background, respectively, relative to the original image. A larger value means that the region left unblurred exerts a stronger influence on the fake prediction. The results show that blurring the foreground leads to consistently larger score increases than blurring the background across all generators (**0.224 vs. 0.080** on average), indicating that the background contributes more to the detector's decision.
>
> | Masked Region | Doubao | Akool | Flux  | Google Imagen | Hailuo | Nano Banana | Stable Diffusion | Mean  |
> | ------------- | ------ | ----- | ----- | ------------- | ------ | ----------- | ---------------- | ----- |
> | FG            | 0.293  | 0.303 | 0.16  | 0.15          | 0.264  | 0.249       | 0.148            | 0.224 |
> | BG            | 0.125  | 0.133 | 0.051 | 0.05          | 0.129  | 0.101       | 0.051            | 0.080 |
>
> **Q3: Analysis on the source of improvement:**
>
> Our experiments show that the performance improvement stems from our peak selection strategy. To validate this, we maintain the same additive calibration architecture but vary how the local patches are selected. Compared to the global baseline (92.1% mAcc), simply adding the logit re-weighting branch with random patches yields only 93.7%, indicating that the structural addition alone provides marginal benefits. Replacing random selection with hard top-scoring selection improves performance to 96.4%, indicating that localizing artifacts is beneficial. Ultimately, our soft peak aggregation achieves the best performance of 98.3%.
>
> | Strategy                         | Midjourney | SD_v1.4 | SD_v1.5 |  ADM  | Glide | Wukong | VQDM  | BigGAN |   mAcc   |
> | :------------------------------- | :--------: | :-----: | :-----: | :---: | :---: | :----: | :---: | :----: | :------: |
> | Global Baseline (No Calibration) |    96.5    |  96.7   |  96.6   | 96.7  | 96.8  |  96.5  | 96.9  |  60.5  |   92.1   |
> | + Random Patches                 |    99.6    |  99.8   |  99.9   | 99.8  | 99.7  |  99.8  | 99.7  |  51.6  |   93.7   |
> | + Hard Top-Scoring Patches       |    96.3    |  96.5   |  96.4   | 96.2  | 96.2  |  96.6  | 96.3  |  96.6  |   96.4   |
> | + Ours (Soft Peak Aggregation)   |   100.0    |  100.0  |  100.0  | 100.0 | 99.9  | 100.0  | 100.0 |  86.7  | **98.3** |

---

> > ### Author Rebuttal · Reviewer_72DQ · 2026-04-04
> >
> > The author claims that the forgery artifaction come from the background rather than the foreground, which does not align with the consensus. I still have doubts about this. Therefore, I decided to keep my score.

---

> > > ### Author Response · Authors · 2026-04-04
> > >
> > > Dear Reviewer 72DQ,
> > >
> > > **Our quantitative evaluations consistently demonstrate that forgery cues in the background are stronger than those in the foreground.**
> > >
> > > To validate this, we performed a **SHAP-based attribution analysis to quantify each region's contribution to the prediction**. By applying SAM-generated masks, we computed the SHAP density for both foreground and background (FG/BG), objectively quantifying which region drives the final prediction.
> > >
> > > The results consistently reveal the same trend across all evaluated generators:
> > >
> > > 1. **The background dominates the positive prediction:** $d^+(\mathrm{BG}) - d^+(\mathrm{FG}) > 0$ in every case (mean: 0.307), indicating the background provides the primary cues for detection.
> > >
> > > 2. **The foreground yields a negative net contribution:** Within the FG itself, positive evidence is weaker than negative evidence, with $d^+(\mathrm{FG}) - d^-(\mathrm{FG}) < 0$ for all models (mean: -0.033), **implying that FG regions often act as distractors rather than reliable fake indicators.**
> > >
> > > These results objectively confirm that models rely primarily on background cues, while foreground contributions are minimal. This supports our premise: generation artifacts tend to be more pronounced and discriminative in the background. We will emphasize this data-driven observation in the revision.
> > >
> > > Best regards,
> > >
> > > Authors of Submission [11378]

---

### Decision · Program_Chairs · 2026-04-30

**Decision:**

Accept (regular)

**Comment:**

Three of the four reviewers have highly appreciated the contributions of this paper and gave positive scores (4, 4, 4). The rest Reviewer 72DQ provided constructive initial comments but did not submit the final justifications, regardless of the AC's reminder. The AC has checked the authors' follow-up replies to Reviewer 72DQ and believes that major concerns have been addressed. In particular, the authors have provided quantified evidence for the concern about the importance of background vs. foreground. However, the reviewer just thought this is inconsistent with "the consensus" without explaining why and where "the consensus"  has been drawn.